

# Daily Landsat-scale evapotranspiration estimation over a forested landscape in North Carolina, USA using multi-satellite data fusion

Yun Yang[1], Martha C. Anderson[1], Feng Gao[1], Christopher R. Hain[2], Kathryn A. Semmens[3], William P. Kustas[1],
Asko Noormets[4], Randolph H. Wynne[5], Valerie A. Thomas[5], Ge Sun[6]

[1]USDA ARS, Hydrology and Remote Sensing Laboratory, Beltsville, Maryland, USA

[2]Earth System Science Interdisciplinary Center, University of Maryland, Maryland, USA

[3]Nurture Nature Center, Easton, Pennsylvania, USA

[4]Department of Forestry and Environmental Resources, North Carolina State University, North Carolina, USA

[5]Department of Forest Resources and Environmental Conservation, Virginia Polytechnic Institute and State University, Virginia, USA

[6]Eastern Forest Environmental Threat Assessment Center, Southern Research Station, USDA Forest Service, Raleigh, North Carolina, USA

*Correspondence to:* Yun Yang. Email:yun.yang@ars.usda.gov

**Abstract.** As a primary flux in the global water cycle, evapotranspiration (ET) connects hydrologic and biological processes and is directly affected by water and land management, land use change and climate variability. Satellite remote sensing provides an effective means for diagnosing ET patterns over heterogeneous landscapes; however, limitations on the spatial and temporal resolution of satellite data, combined with the effects of cloud contamination, constrain the amount of detail that a single satellite can provide. In this study, we

describe an application of a multi-sensor ET data fusion system over a mixed forested/agricultural landscape in North Carolina, USA during the growing season of 2013. The fusion system ingests ET estimates from a Two-Source Energy Balance (TSEB) model applied to thermal infrared remote sensing retrievals of land surface temperature from multiple satellite platforms: hourly geostationary satellite data at 4-km resolution, daily 1-km imagery from the Moderate Resolution Imaging Spectroradiometer (MODIS), and bi-weekly Landsat thermal

data sharpened to 30-m. These multiple datastreams are combined using the Spatial-Temporal Adaptive Reflectance Fusion Model (STARFM) to estimate daily ET at 30-m resolution to investigate seasonal water use behavior at the level of individual forest stands and land cover patches. A new method, also exploiting the STARFM algorithm, is used to fill gaps in the Landsat ET retrievals due to cloud cover and/or the scan-line corrector (SLC) failure on Landsat 7. The retrieved daily ET timeseries agree well with observations at two

AmeriFlux eddy covariance flux tower sites in a managed pine plantation within the modeling domain: US-NC2





located in a mid-rotation (20 year old) loblolly pine stand, and US-NC3 located in a recently clear cut and replanted field site. Root mean square errors (RMSE) for NC2 and NC3 were 0.99 mm d$^{-1}$ and 1.02 mm d$^{-1}$, respectively, with mean absolute errors of approximately 29% at the daily time step, 12% at the monthly time step, and 3% over the full study period at two flux tower sites. Analyses of water use patterns over the plantation

indicate increasing seasonal ET with stand age for young to mid-rotation stands up to 20 years, but little dependence on age for older stands. An accounting of consumptive water use by major land cover classes representative of the modeling domain is presented, as well as relative partitioning of ET between evaporation (E) and transpiration (T) components obtained with the TSEB. The study provides new insights about the effects of forest management and land use change on hydrological water balance, and the method developed has the

potential to be used to routinely monitor hydrology and water use over heterogeneous landscapes using thermal remote sensing data.

## 1 Introduction

Evapotranspiration (ET) is a major component of the water balance and connects hydrologic and biological processes (Hanson et al., 2004; Wilson et al., 2001). ET varies with different climate and vegetation types and is

directly affected by land management strategies and climate change (Pereira et al., 2002). ET is also a key variable in most ecohydrological models and ecosystem service assessments (Abramopoulos et al., 1988; Kannan et al., 2007; Olioso et al., 1999; Tague & Band 2004; Sun et al., 2011). In spite of the importance of ET, routine estimation of ET at high spatial (plot level) and temporal (daily) resolution has not yet been achieved with acceptable accuracy over landscape and regional scales (Wang and Dickinson, 2012).

Current forest ET estimation methods span a range of spatial scales: from individual plants, to tower footprints, to watershed scales (Fang et al., 2015). These methods include in situ measurement, simulation using hydrologic and land surface models which are normally driven by weather data, and estimation from satellite remote sensing data. Techniques for measuring ET include weighing lysimeters (Wullschleger et al., 1998), sap flow (Klein et al., 2014; Smith and Allen, 1996) and plant chambers (Cienciala and Lindroth, 1995), soil water

budgets (Cuenca et al., 1997), eddy covariance (EC; Baldocchi et al., 2001) and catchment water balance (Pan et al., 2012). While EC is a widely used observation method and provides an important data source to many research fields (Baldocchi et al., 2001), it only measures the flux in the footprint area ($10^2$-$10^4$ m$^2$), which is determined by the micro-climate conditions around the flux tower and the instrument height. Catchment water balance is also a frequently used method, calculating ET from long-term precipitation and streamflow



observations with the assumption that the soil water storage change is negligible (Domec et al., 2012; Wilson et al., 2001). All these observation methods have their inherent advantages and limitations, especially when considering both temporal and spatial resolution issues. Another group of forest ET estimation methods is empirically based, establishing a relationship between ET with other parameters; for example, precipitation,

reference ET and vegetation indices (Leaf Area Index (LAI), Normalized Difference Vegetation Index (NDVI) and Enhanced Vegetation Index (EVI)) (Johnson & Trout, 2012; Mutiibwa & Irmak, 2013; Nemani & Running, 1988; Sun et al., 2011; Zhang et al., 2004). Many studies have applied process-based eco-hydrological models to estimate ET (Chen & Dudhia, 2001; Tague & Band, 2004; Tian et al., 2010). These models usually estimate ET from potential ET, which is then regulated by climate data and soil and vegetation characteristics. However, with

the focus on predicting runoff and the soil water profile, studies using hydrologic models generally do not evaluate the performance of ET simulation. To simplify the physical processes, many models assume the plant status is static. This assumption might result in errors in simulating ET dynamics, especially over shorter time periods (seasonally, monthly, weekly or daily) (Méndez-Barroso et al., 2014; Tian et al., 2010). Often physical process-based models involve hundreds of input variables/parameters, many of which are not easily measured or

known in a spatially distributed manner at watershed and regional scales. Although models can be calibrated using local or watershed scale observations, there is the often-mentioned problem of equifinality, where different sets of parameters during calibration give the same simulation results due to the inherent complexity of the system (Beven and Freer, 2001).

Mapping ET using satellite remote sensing data has been widely applied since the 1980s due to growing

interest in the spatial dynamics of water use at the watershed and regional scales (Kalma et al., 2008). Of particular interest in the water resource community are surface energy balance methods based on remotely sensed land-surface temperature (LST) retrieved from thermal infrared (TIR) imagery, which provides proxy information regarding the surface moisture status ( Hain et al., 2011; Anderson et al., 2012a). LST captures signals of crop stress and variable soil evaporation that are often missed by crop coefficient remote sensing

techniques, which are based on empirical regressions with shortwave vegetation indices. Furthermore, diagnostic estimates of ET from the surface energy balance provide an independent estimate of landscape water use that is a valuable benchmark for comparison with estimates based on water balance or hydrologic modeling (Hain et al., 2015; Yilmaz et al., 2014). Finally, the range in spatial resolution and coverage of existing TIR data sources enables mapping of ET from the plot or field scale (<100 m resolution) up to continental or global coverage at 1-

5 km resolution.




The Atmosphere-Land Exchange Inverse model (ALEXI; Anderson et al., 1997, 2007) and associated flux disaggregation algorithm (DisALEXI; Anderson et al., 2004; Norman et al., 2003) are examples of a multi-scale energy balance modeling approach that can utilize LST data from multiple satellite platforms with TIR sensing capabilities. The regional ALEXI model uses time-differential measurements of morning LST rise from geostationary satellites to estimate daily flux patterns at 3-10 km resolution and continental scales. Using higher resolution LST information from polar orbiting systems, DisALEXI enables downscaling of ALEXI fluxes to finer scales, better resolving land-use and moisture patterns over the landscape, and better approximating the spatial scale of ground-based flux observations. Landsat data (30-120 m) can be used to retrieve ET at the field scale, which is particularly useful for water management applications. However, due to the lengthy revisit interval (8 to 16 days), further lengthened by cloud contamination, the number of useful Landsat scenes that can be acquired during a growing season is limited. The Moderate Resolution Imaging Spectroradiometer (MODIS) has a shorter revisit interval (approximately daily) but is too coarse (1 km in the TIR bands) for field-scale ET estimation. Cammalleri et al.(2013) proposed a data fusion method to combine ET estimates derived from geostationary, MODIS and Landsat TIR data, attempting to exploit the spatiotemporal advantages of each class of satellite to map daily ET at a sub-field scale. This ET fusion approach has been successfully applied over rain-fed and irrigated corn, soybean and cotton fields (Cammalleri et al., 2014), as well as irrigated vineyards (Semmens et al., 2015). The work described here constitutes the first application to forest land cover types, representing a substantially different roughness and physiological regime than that of shorter crops.

In this paper, ALEXI and DisALEXI are applied over a commercially managed loblolly pine (*Pinus Taeda*) plantation, representing a range in stand age, to estimate daily field scale ET using the data fusion methodology. Retrieved 30-m ET timeseries are evaluated at two flux tower sites, sited in mature and recently clear-cut pine stands. The objective is to (1) study how well ALEXI and DisALEXI can be used to estimate ET over forested sites; (2) evaluate the models' ability to capture the dynamics of fluxes over the contrasting canopy structures in both pine and the adjacent vegetation; and (3) investigate the utility of daily field-scale ET retrievals for water resource management in forested systems. We also present a new method, based on data fusion, for filling gaps in Landsat-based ET retrievals due to partial cloud cover as well as the scan-line corrector (SLC) failure in Landsat 7 imagery.

## 2 Methods

### 2.1 Thermal-based multi-scale ET retrieval



The regional Atmosphere-Land Exchange Inverse (ALEXI) and the associated flux disaggregation model (DisALEXI) are based on the Two Source Energy Balance (TSEB) land-surface representation of Norman et al. (1995), with further refinements by Kustas and Norman (1999, 2000). Rather than treating the land-surface as a homogeneous surface, TSEB partitions modeled surface fluxes and observed directional radiometric surface temperature between soil and vegetation components:

$$T_{RAD}(\emptyset)^4 = f(\emptyset)T_c^{\,4} + [1 - f(\emptyset)]T_s^{\,4} \qquad (1)$$

where $\emptyset$ is the thermal view angle, $f(\emptyset)$ is the fractional vegetation cover apparent at the thermal view angle, $T_{RAD}$ is the directional radiometric temperature, $T_c$ is the canopy temperature, and $T_s$ is the soil temperature. The surface energy balance for the canopy, soil and combined system is represented in Eq. (2):

$$RN = \lambda E + H + G$$

$$RN_s = H_s + \lambda E_s + G$$

$$RN_c = H_c + \lambda E_c \qquad (2)$$

where the subscripts "$C$" and "$S$" represent fluxes from the canopy and soil components; and $RN$ is net radiation, $\lambda E$ is latent heat flux, $H$ is sensible heat flux and $G$ is soil heat flux. Component surface temperatures in Eq. 1 are used to constrain $RN$, $H$ and $G$; canopy transpiration ($\lambda E_c$) is initially estimated with a modified Priestley-Taylor approach under the unstressed conditions assumption, and then iteratively down regulated if $T_c$ indicates canopy stress, ruled by the assumption that condensation under daytime clear-sky conditions is unlikely; while soil evaporation ($\lambda E_s$) is computed as a residual to the soil energy budget. Further information regarding the TSEB model formulation is provided by Kustas and Anderson (2009).

Roughness length ($Z_m$) impacts the aerodynamic resistance ($R_a$), which is the resistance to heat transport across the layer between the nominal heat exchange surface within the canopy and the air temperature measured height. The aerodynamic resistance ($R_a$) can be expressed as Eq. (3) (Brutsaert, 1982):

$$R_a = \frac{\left[\ln\left(\frac{Z_T - d}{Z_m}\right) - \psi_h\right]\left[\ln\left(\frac{Z_u - d}{Z_m}\right) - \psi_m\right]}{k^2 u} + R_{ex} \qquad (3)$$

where $k$ is the von Karman constant (0.4); $u$ is the wind speed measured at height $Z_u$; $Z_T$ is the air temperature measured height; $d$ is the displacement height; $Z_m$ is the roughness length, which can be estimated from the nominal canopy height $(h_c)$, $Z_m \approx {h_c}/{8}$ (Shaw and Pereira, 1982); $\psi_h$ is the stability corrections for heat transport; $\psi_m$ is the stability corrections for momentum transport; and $R_{ex}$ is the excess aerodynamic resistance.



The regional scale ALEXI model applies the TSEB in time-differential mode using measurements of morning LST rise obtained from geostationary platforms (Anderson et al. 1997; Anderson et al. 2007). Energy closure over this morning period is obtained by coupling the TSEB with a simple model of atmospheric boundary layer (ABL) development (Fig. 1). This time-differential approach reduces model sensitivity to errors in LST

retrieval due to atmospheric and surface emissivity effects, but it does constrain ALEXI ET estimates to the relatively coarse spatial scales typical of geostationary satellites. To estimate ET at the finer scales required for many management applications, the ALEXI fluxes can be spatially disaggregated using the DisALEXI approach (Anderson et al., 2004; Norman et al., 2003). DisALEXI uses, as an initial estimate, air temperature estimates diagnosed by ALEXI at a nominal blending height at the interface between the TSEB and ABL submodels, along

with high spatial resolution images of surface temperature data and vegetation cover fraction from polar orbiting or airborne systems, to run the TSEB at sub-pixel scales over each ALEXI pixel area. The TSEB fluxes are reaggregated and compared with the ALEXI pixel flux, and the air temperature boundary condition is iteratively modified until the fluxes are consistent at the ALEXI pixel scale. More details on the ALEXI/DisALEXI multi-scale modeling system can be found in Anderson et al., 2004, 2011and 2012b.

**2. 2 Data processing and fusion system**

ET retrievals generated with DisALEXI using TIR data from MODIS (near daily, at 1 km resolution) and Landsat (periodic, sharpened to 30 m resolution) are gap-filled and then fused into a daily time series at 30 m. The major components of the processing stream are described in greater detail below, including a Data Mining Sharpener (DMS; Gao et al., 2012b) tool that is used to improve the spatial resolution of the LST inputs to DisALEXI; the

Spatial and Temporal Adaptive Reflectance Fusion Model (STARFM; Gao et al. 2006), which is used to combine temporally sparse Landsat and dense MODIS ET maps to produce daily Landsat-scale ET time series; and a gap-filling procedure that is applied to ALEXI and MODIS and Landsat DisALEXI retrievals prior to disaggregation and fusion. The gap-fill and fusion processes are schematically represented in Fig. 2.

**2.2.1 Data Mining Sharpener (DMS)**

In both the Landsat and MODIS imaging systems, the TIR sensors have significantly lower spatial resolution than the shortwave instruments on the same platform. For Landsat, TIR resolution varies from 60 m (Landsat 7) to 100 m (Landsat 8) to 120 m (Landsat 5), while the shortwave images are processed to 30 m. For MODIS, TIR resolution is 1km while the shortwave resolution is 250 m. Particularly for Landsat, there is benefit to mapping



ET at 30 m rather than the native TIR resolution, as boundaries in land cover and moisture variability are much better defined.

To enable higher resolution ET mapping, the DMS sharpening tool is implemented within the ET fusion package. The DMS technique creates regression trees between TIR band brightness temperatures and shortwave spectral reflectance from both moving windows and the whole scene (Gao et al., 2012b). The original TIR data are sharpened from their native spatial resolution to finer resolution with DMS, with the choice of using all of the available shortwave bands or a subset of these bands or even a single vegetation index as input to the regression tree. The sharpened results from the regression tree created by selecting samples locally from an overlapped moving window are then combined with the results from the regression tree created from the whole scene based on a weighted factor, calculated from the residuals of the two sharpened results.

### 2.2.2 Spatial and Temporal Adaptive Reflectance Fusion Model (STARFM)

The STARFM algorithm combines the spatial information from Landsat imagery with temporal information from the coarser but more frequently collected MODIS imagery to produce daily estimates at Landsat-like scale. STARFM was originally designed to fuse shortwave reflectance data, but has demonstrated utility in fusing higher order satellite products as well, as long as there is sufficient consistency between the Landsat and MODIS retrievals.

First, ET data from both MODIS and Landsat retrievals are extracted onto a common 30-m grid. A moving searching window method is then used in STARFM to estimate values at the center pixel of the moving window.

$$L\left(x_{p/2}, y_{p/2}, t_0\right) = \sum_{i=1}^{p} \sum_{j=1}^{p} \sum_{k=1}^{n} W_{ijk} \times \left( M\left(x_i, y_j, t_0\right) + L\left(x_i, y_j, t_k\right) - M\left(x_i, y_j, t_k\right) \right) \quad (4)$$

where p is the size of the moving window and $\left(x_{p/2}, y_{p/2}\right)$ is the center pixel of the moving window that needs to be estimated at time $t_0$. $\left(x_i, y_j\right)$ is the pixel location, $M\left(x_i, y_j, t_0\right)$ is the MODIS pixel value at time $t_0$, $M\left(x_i, y_j, t_k\right)$ is the MODIS pixel value at time $t_k$ and $L\left(x_i, y_j, t_k\right)$ is the Landsat pixel value at time $t_k$. $W_{ijk}$ is the weighting factor that determines how much each pixel in the moving window contributes to the estimation of the center pixel value. In this study, STARFM uses the weighting function derived from Landsat ET and MODIS ET retrieved on the same date and MODIS ET on the prediction date to get Landsat-like ET estimations on all prediction dates between Landsat overpasses.

### 2.2.3 ET Gap-filling Methods





Spatiotemporal gaps in TIR-based ET retrievals occur for a variety of reasons, including cloud cover, frequency of sensor overpass, limitations imposed to avoid distortions in LST data acquired at large off-nadir view angles, and other sensor issues. Prior to disaggregation and fusion, the input ET fields have been gap-filled, both spatially and temporally, to the extent possible to ensure relatively gap-free output time series.

Due to the high temporal frequency of data acquisition from both geostationary and MODIS systems, the ALEXI and DisALEXI-MODIS retrievals can be reasonably gap-filled and interpolated to daily time steps in all but the cloudiest of circumstances. Time intervals between clear-sky Landsat acquisitions are too lengthy in general, motivating the need for data fusion to fill temporal gaps. Spatial gaps in Landsat ET retrievals have been filled using a method based on STARFM, as described below.

*ALEXI and MODIS-DisALEXI*

Gaps in the daily ET maps from ALEXI and MODIS-DisALEXI were filled using the method described by Anderson et al. (2012b). Daily reference ET is first calculated using the Food and Agriculture Organization (FAO) Penman-Monteith formulation for a grass reference site (Allen et al., 1998). The ratio of actual-to-reference ET ($f_{RET}$) is computed and then filtered, smoothed and gap-filled at each pixel using a Savitzky-Golay

filter. Gap-filled daily ET is recovered by multiplying this $f_{RET}$ series by daily reference ET.

*Landsat-DisALEXI*

To ensure optimal spatial coverage in the fused 30 m daily time series, the Landsat-based ET retrievals on Landsat overpass dates must also be gap-filled to the extent possible. Gaps in Landsat ET result from cloud cover, or in the case of Landsat 7, missing pixels due to the SLC failure that occurred since May 2003, resulting

in striped gaps in all but the center of each scene. In the case of Landsat, the gap between usable overpasses may be too long to justifiably use the $f_{RET}$ approach applied to the ALEXI and MODIS time series. Therefore, an alternate method has been developed to fill cloud-gaps/stripes to create filled scenes for ingestion into STARFM.

     The method involves running STARFM for the partly cloudy or striped prediction date using Landsat retrieved ET from surrounding clear dates. The cloud/stripe-impacted areas in the Landsat retrieval are then filled

as a weighted function of the STARFM estimated Landsat-like ET and the Landsat retrieved ET. This weighting is implemented to reduce impacts of bias that may exist between the STARFM estimate and the actual retrieval in the area of the gap, which could otherwise result in a notably patchy fill. The weighting function is computed within a moving window, predicting ET at the center pixel. The weighting value of each pixel in the moving window is calculated based on land cover type, spatial distance to the center predicting pixel and pixel value and

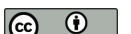



is then normalized to a 0-1 value. Pixels that have the same land cover type as the predicting pixel, a short spatial distance to the predicting pixel and less difference in pixel values, have higher weighting score. The resulting filled value is computed as

$$FilledValue = A_L - A_S + \sum_i^p \sum_j^p (W_{i,j} \times S_{i,j}) \tag{5}$$

where $A_L$ is the average of pixels in the moving window in Landsat retrieved ET, $A_S$ is the average of pixels in the moving window in STARFM fused ET at the same day of landsat retrieved ET, i and j is the pixel location in the moving window, $p$ is the moving window size, $W$ is the weighting score and $S$ is the STARFM value.

The searching distance is predefined based on the heterogeneous condition of the study area. A larger searching distance normally requires longer computing time and might result in more random noise. Searching distance that is too small might not be able to provide enough similar surrounding pixels to predict the value of the center pixel. As described in the weighting scheme described above, pixels that are far away from the center pixel have lower weighting than pixels that are close to the center pixel. When the gap area is large and contiguous (more than 80% of the moving window), there are not enough good pixels that can provide useful information for the gap-fill. In this case, there is no gap filling.

The cloud mask used in this study is the Fmask (Function of mask) data from the Level 2 surface reflectance product distributed by EROS (Earth Resources Observation and Science) center. Fmask uses Landsat Top of Atmosphere (TOP) reflectance and brightness temperature as inputs to produce cloud, cloud shadow, water and snow mask for Landsat images (Zhu and Woodcock, 2012). Cloud physical properties are first used to produce potential cloud pixels and clear-sky pixels and normalized temperature probability, spectral variability probability and brightness probability are combined to estimate cloudy area. The cloud shadow area is derived from the darkening effect of the cloud shadows in the near-infrared band, view angle of the satellite sensor and the illuminating angle. In this study, we flagged class 2 (cloud_shadow) and 4 (cloud) in the cloud mask file as cloud.

## 3 Experimental site and datasets

### 3.1 Study area

The study area (Fig. 3) is located over the Parker Tract in the lower coastal plain of North Carolina. The Parker Tract consists of loblolly pine plantations of different ages and native hardwood forests (Noormets et al., 2010). The loblolly pine plantations are commercially managed for timber production by Weyerhaeuser Company. The study area is flat, about 3 m above sea level, and has been ditched (4[th] order ditches at 100m spacing) to manage



the water table and improve tree productivity (Domec et al., 2012). The soil is Belhaven series histosol, with 50-85 cm organic layer over coarse glacial outwash sand (Sun et al., 2010). The study area is classified as outer coastal plain mixed forest province (Bailey, 1995). The long-term (1945-2008) monthly temperature ranges between 26.6 ℃ in July to 6.4 ℃ in January, with an annual mean temperature of 15.5 ℃. The long-term annual

precipitation is around 1320 ± 211mm, relatively evenly distributed throughout the year.

Evaluation of the DisALEXI ET estimates was performed at two AmeriFlux tower sites in this area: US-NC2 (35º48´N, 76º40´W) and US-NC3 (35º48´N, 76º39´W). US-NC2 is a mid-rotation plantation stand with 90 ha area, which was established after clearcutting a previous rotation of loblolly pine, replanted with 2-year old seedlings at 1.5m by 4.5m spacing in 1992. The stand has been fertilized twice – at establishment, and in 2010,

following a thinning in 2009. The tree density at the time of the current study in 2013 was 171 trees per hectare and standing biomass of 42.6 t C ha$^{-1}$ in the overstory and 6.5 t C ha$^{-1}$ in the understory. The understory was composed of red maple, greenbrier and volunteer loblolly pine. US-NC3 was established in 2013 in a stand that was clearcut in 2012, located approximately 1.5 km from US-NC2. US-NC3 was replanted with seedling loblolly pines after the clearcut. US-NC2 site was 22 years old in 2013, 19.0 m tall, and had a mean LAI of 3.77 m$^2$ m$^{-2}$,

whereas NC3 was freshly planted with 2-year old seedlings, 0.2 m tall, and had no overstory leaf area. The mostly herbaceous understory contained 85±52 g C m$^{-2}$ at NC3.

Both NC flux towers are equipped with similar instrumentation, and biophysical data are collected routinely. These measurements are described below.  This study focuses on data collected during the 2013 growing season, starting after the launch of Landsat 8 on February 11, 2013 and continuing until November 8, 2013.

**3.2 Micrometeorological and land management data**

At both NC2 and NC3, energy fluxes were measured using an open-path eddy covariance system, which includes a CSAT3 three-dimensional sonic anemometer (Campbell Scientific Instrument-CSI, Logan, UT, USA[1]), a CR5000 data logger (CSI), an infrared gas analyzer (IRGA, Model LI-7500, LI-COR, Lincoln, NE, USA) and a relative humidity and air temperature sensor (model HMP-45C; Vaisala Oyj, Helsinki, Finland) (Sun et al.,

2010). Soil heat flux was measured at NC2 with three heat flux plates (model HFT3, CSI, Logan, UT, USA) at the depth of 2cm. The 3 soil heat flux plates were placed in three contrasting microsites - one in a row of trees, in relative shade, another between rows in a mostly open environment and one about half-way in-between.

---

[1] The use of trade, firm, or corporation names in this article is for the information and convenience of the reader. Such use does not constitute an official endorsement or approval by the United States Department of Agriculture or the Agricultural Research Service of any product or service to the exclusion of others that may be suitable.





Measurements of G at NC3 site are not available for 2013 due to an instrument failure. Net radiation was measured with 4-component net radiometers (Kipp & Zonen CNR-1, Delft, Netherlands) at each of the two towers. Precipitation was measured by two tipping bucket type of rain gages (TE-525, CSI; Onset Data Logging Rain Gauge, Onset Computer Corporation, USA).

Flux observations at 30-min time steps were quality checked, as judged by atmospheric stability and flux stationarity (Noormets et al., 2008). The 30-min data were then gap-filled using the monthly regression between observed and potential ET models created from good quality observed data. The energy imbalance problem was checked and the average closure ratio of 30-min dataset at NC2 site was 0.88 during daytime when net radiation is larger than 0. Since there was no soil heat flux observation at NC3 site, there was no closure information and

the observed latent heat was used to compare with the simulated data. The 30-min energy fluxes during the daytime were summed up to get daily energy fluxes for validation.

Stand age maps and tree planting history for the study area were obtained from the Weyerhaeuser Company. The stand age ranges from 1 to 89 years, with most stands under 30 years of age. Since this information is proprietary, the stand age maps cannot be displayed; however, these data were used statistically to assess the

relationships between water use and stand age. A 60 m buffer inside the edge of each field was applied to exclude the pixels mixed with roads or other fields. All the other pixels were used to assess the relationships between water use and stand age.

### 3.3 ALEXI/DisALEXI model inputs

The ET estimation process involves fusion of data from three major geostationary and polar orbiting satellite

systems: GOES, MODIS and Landsat.  In addition, each ET retrieval pulled meteorological inputs (air temperature, wind speed, vapor pressure, atmospheric pressure and insolation) from a common gridded dataset, generated at hourly time steps and relatively coarse spatial resolution (32 km) as part of the North American Regional Reanalysis (NARR).

Land-surface temperature (LST) data from the GOES Imager instruments were used to run ALEXI over the

continental U.S. for 2013 at 4 km resolution (Anderson et al., 2007).  In addition, MODIS (4-day) LAI products (MCD15A3) were aggregated from 1 to 4-km and interpolated to daily using the smoothing algorithm developed by Gao et al. (2008). LAI is used in a Beer's Law formulation to estimate $f(\emptyset)$ for Eq. 1, and to assign land cover class dependent vegetation heights for roughness parameterization  (See Anderson, et al., 2007).

MODIS products used in the MODIS disaggregation include instantaneous swath LST (MOD11_L2; Wan et

al.,  2004), geolocation data (MOD03), NDVI (MOD13A2; Huete et al., 2002), LAI (MCD15A3, Myneni et al.,


2002), albedo (MCD43GF, Schaaf et al., 2011), and land cover (MCD12Q1, Friedl et al., 2002). The LST swath data, at 1 km spatial resolution, was converted to geographic coordinates using the IDL-based MODIS reprojection tool. The NDVI product (1 km spatial resolution) is produced at 16 day intervals, LAI (1 km) at 4 days, and albedo (1 km) at 8 days. All data were quality checked using a data quality filter. The MODIS NDVI, LAI and albedo data were bilinearly interpolated to estimate daily values. MODIS LST was sharpened using NDVI to reduce the off-nadir pixel smearing effect.

Landsat 8 thermal infrared and shortwave surface reflectance data from 2013 used to run DisALEXI were obtained from USGS. Eight relatively cloud-free (> 75%) Landsat scenes (path 14 and row 35) were available during the 2013 growing season, including 1 Landsat 7 scene and 7 from Landsat 8 (Table 1). Landsat-scale LAI was retrieved from Landsat shortwave surface reflectance data using MODIS LAI products as reference (Gao et al., 2012a). LST was sharpened to 30m using blue, green, red, near infrared, SWIR1 and SWIR2 bands (refer to sec 2.2.1 for more details about Landsat LST sharpening).

Land cover type was used in both Landsat and MODIS disaggregations to set pixel-based vegetation parameters including seasonal maximum and minimum vegetation height (used in the surface roughness formulations), leaf size and leaf absorptivity in the visible, NIR and TIR bands following Cammalleri et al. (2013). For DisALEXI using Landsat, the 30 m National Land Cover Dataset (NLCD) for 2006 developed by University of Maryland (UMD) (Wickham et al., 2013) was used. For the MODIS disaggregation, the UMD land cover datasets was resampled to 1 km resolution using the dominant class in each pixel.

## 4 Results

### 4.1 Performance of the Landsat gap-filling algorithm

Examples of results from Landsat gap-filling method are shown in Fig. 4 and Fig. 5. For DOY 96 (Fig. 4), L7 SLC stripes and also a few cloudy areas were filled by combining the direct Landsat retrieval (left panel) with the STARFM ET prediction for DOY 96 using a Landsat-MODIS image pair from DOY 104. The cloudy areas in DOY 200 (Fig. 5) were filled using a Landsat-MODIS pair from DOY 152. In each case, the size of the overlapped moving window was 420 m by 420 m. This means that contiguous gaps larger than the window were not filled since there were not enough candidate pixels to create the statistical relationship needed between the direct Landsat retrieval and STARFM ET. The white rectangular box located in the northeast area of both figures contains no data because more than 40% of Landsat pixels within the 4-km ALEXI ET pixel were affected by a large water body.

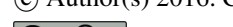



In each of these cases, the spatial patterns within missing regions in the direct retrievals appear reasonably reconstructed with the ET gap-filling method, with no obviously patchy artifacts in the gap-filled ET images. Even linear structures were restored, for example roads and field boundaries. The gap-filling method works even better when the small or linear objects are very different from surrounding pixels. The choice of moving window

size can affect the gap-filling results and may require scene-based adjustment, with the need to balance the risk of inappropriate candidate pixel selection if the window is too large, or with a lack of candidate pixels if the window is too small.

Figure 6 shows a synthetic study used to assess the accuracy of the gap-filling procedure. Here, a direct Landsat ET retrieval for DOY 152 was artificially masked using SLC stripes from DOY 96. The final panel

shows the gap-filled image. Comparing the original values with the gap-filled values yields an $R^2$ of 0.89 and MAE is -0.01 mm d$^{-1}$, with the average of original values as 5.81 mm d$^{-1}$ and the average of gap-filled values as 5.80 mm d$^{-1}$.

The gap-filling method relies on the inputs from both the original Landsat ET and the STARFM prediction, which in turn relies on a filled MODIS image on the target date as well as a MODIS-Landsat image pair on a

surrounding date. If the Landsat image in the input pair also has gaps, additional pairs can be used to iteratively fill the target image. Chen et al. (2011) applied a similar weighting function in a moving window to fill the Landsat 7 SLC-off images using an appropriate TM image or SLC-on ETM+ image. Roy et al. (2008) used both MODIS BRDF/Albedo product and Landsat observations to predict Landsat reflectance with a semi-physical fusion approach. In comparison with earlier studies concerned with gap-filling of Landsat 7 SLC-off images, the

current method used both MODIS retrieved ET and Landsat retrieved ET to gap-fill the missing ET while the other studies focused on gap-filling the spectral reflectance.

## 4.2. Evaluation of Daily ET Retrievals from DisALEXI at the Flux Tower Sites

Modeled and measured instantaneous and daytime integrated surface energy fluxes on Landsat overpass dates are compared in Fig. 7, demonstrating good correspondence. Statistical performance metrics for each flux component

for both sites are shown in Table 2, including mean absolute error (MAE), root mean square error (RMSE) and mean bias error (MBE). The model performance for each flux is similar between sites, with somewhat lower errors obtained for the clear-cut site (NC3). The latent heat/ET observed at the NC2 site is higher than that at NC3 with or without closure enforcement. As mentioned earlier, closure could not be assessed at NC3 due to failure among the soil heat flux instrumentation. At NC2, closure by residual resulted in an increase in observed

ET by approximately 12% on average.



Timeseries of ALEXI ET (4-km), Landsat ET retrieved on Landsat overpass dates and Landsat-MODIS fused ET (both at 30-m resolution) are compared in Fig. 8 with observed ET 3 site from DOY 50 - 330 for both the NC2 and NC3. In addition, daily ET values generated using a simple Landsat-only interpolation scheme are shown for comparison. These were generated using the MODIS and ALEXI gap-filling technique described in Sec 2.2, conserving the ratio of actual-to-reference ET between Landsat overpass dates. Metrics of statistical performance are listed in Table 3.

Figure 8 highlights the value of disaggregation to the tower footprint scale for model. For NC2, ALEXI 4-km fluxes agree well with tower observations, suggesting that the tower footprint at NC2 is reasonably representative of the surrounding 4 km ALEXI pixel area. The disaggregated 30 m fluxes are also similar to both ALEXI and observations at this site. At NC3, however, the 4-km ALEXI fluxes are notably higher than the observed ET, while the disaggregated fluxes are comparable. The NC3 tower site is not representative at the ALEXI pixel scale, and disaggregation to the tower footprint scale is required to account for local sub-pixel heterogeneity. Even at 1 km resolution, the MODIS retrieval accuracy was degraded at NC3 in comparison with the Landsat-scale retrievals (Table 3). Recall that NC3 was recently clearcut with surrounding areas still comprised by more mature forest stands. With the 4-km spatial resolution, ALEXI ET is able to capture the ET status of the major land cover type (i.e., mature forest), but not the particular patch of land where the NC3 tower is located. This underscores the need for appropriate spatial resolution when comparing modeled with observed fluxes, especially for the more heterogeneous land surfaces (e.g., Anderson et al., 2004).

Overall, the performance of the two Landsat retrievals (STARFM and Landsat-only) are comparable between sites, with RMSE at daily time steps of ~ 0.8 to 1.0 mm d$^{-1}$, and MAE of 0.6 to 0.8 mm d$^{-1}$ (19-30% of the mean observed ET). At monthly time steps, performance improves to 11% - 14%, due to averaging of random errors – including errors in daily insolation forcings from the NARR meteorological dataset. Fluxes are somewhat underestimated at the end of the growing season at each site due largely to the Landsat retrieval on DOY 312. This highlights the importance of good temporal sampling at the Landsat scale – an additional Landsat scene around DOY 270 during the prolonged gap in coverage may have improved the seasonal water use estimates. A small negative mean model bias is observed for both sites, due primarily to underestimation at the end of growing season of the Landsat retrieval on DOY 312. This is also the reason that Landsat-only interpolated ET performs slightly better than STARFM. More details about the comparison of the two Landsat retrievals can be found in the discussion section.

The seasonal cumulative ET at NC2 and NC3 calculated from both the observed ET and the Landsat-MODIS fused timeseries is shown in Fig. 9. There are small divergences between day 120 and day 150 and between day





210 and day 270. Over the periods of accumulation at each site, the error in modeled cumulative ET was 3% of the total observed flux at NC2 and -4% at NC3. Overall, the modeled and measured cumulative ET curves agree well throughout the growing season, indicating the remote sensing method has utility for water use management and assessment at sub-seasonal timescales.

## 4.3 Spatiotemporal variability in seasonal water use

### 4.3.1 ET Variations with Land Cover

Figure 10 shows the land cover types over the study area as described in the NLCD from 2006. [Note: an updated NLCD map for 2011 was published after the study was implemented, but there was no notable change in land cover types over the study area in comparison with NLCD 2006.] The major classes represented in NLCD over the study area include crop land (including corn, cotton and soybeans), forest and woody wetland. The land cover in many plots within the Parker Tract plantation, including the NC2 site, was classified as woody wetland rather than evergreen forest. This misclassification, however, had little impact on the model ET estimates at NC2 due in part to the normalization constraint imposed by the 4 km ALEXI ET output.

Timeseries maps of monthly and cumulative ET in Fig. 11 over the study area exhibit spatiotemporal water use patterns that are related to land cover type (Fig. 10). The relatively high rates of ET during midseason in the riparian and more densely forested regions are readily apparent. Water use patterns in the cultivated agricultural areas reflect the diversity of crops and water management strategies. Within the Parker Tract plantation, a few fields with persistently low ET may be fresh clear-cuts, possibly with a layer of slash to inhibit emergence of new vegetation. In the summer of 2014, after the slash has been collected and pile, these plots may appear more like the recent clearcut near NC3.

Seasonal ET timeseries were developed for five generalized land cover classes (crop land, natural forest, woody wetland, mature plantation and young plantation) to assess variability in water use with landuse/landcover type in the study area. The term "natural forest" is used to describe unmanaged mixed forested areas within the study domain. "Mature plantation" refers to managed stands of loblolly pine within the Parker Tract with ages ranging from 10–20 years, while "young plantation" indicates stand ages less than 3 years. Figure 12 shows the timeseries of modeled field scale ET averaged from 10 randomly sampled pixels associated with each generalized land cover class for 2013. For the woody wetland class, care was taken to select pixels that were correctly classified by visual inspection of Google Earth imagery. A seven-day moving average was applied to the modeled daily ET to reduce noise and facilitate visual comparison.





Pixels classified as woody wetland, natural forest, agriculture and mature plantation generally showed higher ET than did the young plantation pixels. Water use in the woody wetland areas was the highest among all the different land cover types, but was similar to natural forest and mature managed forest during the peak growing season. The seasonal cumulative ET from these five land cover types is shown in Fig. 13. The woody wetlands

tended to have higher seasonal ET than the other three classes, slightly exceeding that of natural forest and mature pine plantations. Modeled water use in crop lands exceeded that in young plantation stands, resulting from relatively higher LAI and lower LST observed over the cropped areas.

Figure 14 shows the average cumulative ET between DOY 50 and 330 associated with the five different land cover types, computed from the 10 random samples per class, and the black bars represent the standard deviation

among the samples. Natural forests showed the lowest variability in ET (30 mm), while the woody wetland and mature plantation pixels had the highest standard deviations (74 and 73 mm, respectively. The high variability in the latter classes may reflect both management effects and misclassification. Crop lands and young forest plantations showed moderate variability in water use, with standard deviations of 64 and 50 mm, respectively. In terms of coefficient of variation in water use across the modeling domain, the crops and young plantation classes

are relatively high at 7.1%, compared to natural forest at 2.8%. Crops and young plantations also have higher coefficient of variation in LAI than other land cover types, leading to larger variability in water use demand through transpiration.

Because the two-source land-surface representation in DisALEXI also provides estimates of the evaporation (E) and transpiration (T) components of ET, the model output can also be used to assess variability in E/T

partitioning between landcover types and through the season (Fig. 15). In general, soil evaporation losses account for a higher percentage of total ET early in the season, after the spring rains but before the canopies have completely leafed out. On average through the season, the E/T ratio is highest for crop lands, young plantations and woody wetlands, which is reasonable given the lower leaf area characteristic of these classes, and the abundant substrate moisture in the case of woody wetlands. Partitioning to T is maximized during peak growing

season (DOY 152, 200 and 248) for all the land cover types. Natural forests and mature plantations tend to have higher transpiration than other land cover types.

### 4.3.2 ET variations with LAI

Many forest hydrology models (Lu et al., 2003; Scott et al., 2006; Sun et al., 2011a) assume seasonal ET is well-correlated with LAI. This assumption was tested over the randomly chosen points from different land cover

types. Sample points from mature plantations, young plantations and crop lands were all located in drained areas,





while sample points from natural forest and woody wetland were in un-drained areas. However, LAI values from the sample points were not affected by drainage conditions. High LAI values were from both drained mature plantations and un-drained natural forest, while relatively low LAI values were from both drained young plantations and un-drained woody wetlands. Figure 16 shows that some, but not all variability in ET as predicted by the fusion estimates is explained by variability in average LAI over the prediction time period ($R^2$=0.59). These results indicate that in 2013 increasing LAI added approximately 350 mm of seasonal water use on top of nearly 800 mm from the soil evaporation contribution. This is reasonable, given that the study area was fairly wet during 2013 due to plenty of precipitation and shallow ground water tables.

### 4.3.3 ET variations with plantation stand age

Within the managed pine plantation at Parker Tract, we also examined variations in seasonal water use with stand age (Fig. 17). This has relevance to forest management practices and their impacts on the water yield of the watershed, which is the difference between precipitation and evapotranspiration over the long term. Higher water yield translates to higher streamflow available for downstream use. Many forest management practices, for example thinning and reforestation, need to consider the influence of stand age on the hydrological response.

Plotting cumulative ET at DOY 330 from various sites against stand age (Fig. 17), there is a clearly positive linear relationship between water use and stand age for the younger stands, between a few years old till around 20 years ($R^2$ is 0.82). As the stand age increases, more water is used as expected to sustain larger amounts of biomass. Differences in seasonal cumulative ET curves for different stand ages begin to significantly diverge after DOY 130 - around the middle of May (not shown). When the stand matures beyond 20 years, the water usage tends to plateau and may actually decrease slightly for stands with trees older than 75 years

## 5 Discussion

### 5.1 Utility of TIR-based data fusion as a daily ET estimation method

In this study, the STARFM modeling approach resulted in a relative error in ET at the daily time step of 28% for the mid-rotation pine plantation site and 31% for the clear-cut site, and errors of 13.6% at the monthly time step. The accuracy of these results is comparable with earlier studies applying the STARFM ET data fusion approach. Over rainfed and irrigated corn and soybean sites in central Iowa, STARFM yielded a relative error of about 11% over eight flux towers (Cammalleri et al., 2013). When STARFM was applied to the Bushland, TX, the daily ET estimation had a relative error of 26.6% for irrigated area and 27% for rainfed area (Cammalleri et al., 2014). For a study area near Mead, NE, the relative error is 20.8% for irrigated crops and 25.4% for rainfed crops



(Cammalleri et al., 2014). Semmens et al. (2015) estimated daily Landsat scale ET over California vineyards with relative error of 18% for an 8-year and 23% for a 5-year vineyard.

Cammalleri et al. (2013) obtained better performance with STARFM over the rainfed agricultural sites studied in central IA in comparison with a Landsat-only interpolation scheme. By fusing MODIS, effects of a rainfall event that occurred between Landsat overpasses were captured as a period of enhanced ET. In contrast, in the current study STARFM yielded marginally lower accuracy than the Landsat-only interpolation. Several factors may be influencing the difference in performance in these two circumstances.

One factor may be differences in climate and moisture status between the two experiments. During the first part of Soil Moisture Experiments in 2002 (SMEX02), conditions were becoming quite dry and the crops – particularly the corn fields – were becoming notably stressed, exhibiting leaf curl near the field edges. A rainfall event near the beginning of July (between Landsat overpasses) significantly relieved the crop stress, and greatly impacted soil moisture conditions over the SMEX02 study area (Anderson et al., 2013). In contrast, the climate of the Parker Tract study site studied here was very wet during 2013, and vegetation condition was not water limited. Consequently, as long as other factors do not affect the health of the vegetation (e.g., pest/disease infestation) or large variations in atmospheric demand (large oscillations in radiation, wind and temperature) the evapotranspiration process will remain near a constant fraction of potential and may be reasonably captured by a simple daily interpolation scheme. There were no major changes in soil moisture status that were additionally captured by the MODIS ET retrievals. Another factor may be that the main and subdominant vegetation type of this study site are trees, which generally have deeper roots than crops and can extract water available much deeper in the soil profile. This further reduces variability in vegetation response during the study period, in comparison with the SMEX02 study.

These two factors are consistent with the assumptions in the Landsat-only interpolation, which assumes the ratio between actual ET and reference ET is constant over time. However, for rainfed cropland areas (like the central Iowa sites) that occasionally go through relatively dry and wet periods, STARFM may be better able to capture time variability in $f_{RET}$ in response to changing moisture conditions.

## 5.2 Comparison with prior ET studies over the Parker Tract

Direct measurements of ET and its components at the study area are only available for pine plantations (NC2 and NC3 AmeriFlux sites) (Sun et al., 2010; Domec et al., 2012). Reported annual ET rates for the NC2 site by Sun et al. (2010) vary from 892 mm (a dry year, 2007)) to 1226 mm in a normal year (2006). A process-based forest hydrological model (DRAINMOD-FOREST) has been calibrated for the NC2 eddy flux site using drainage and





groundwater table data for the period of 2005-2012. ET estimates by the calibrated model vary from 903 mm year$^{-1}$ in 2008 to 1170 mm year$^{-1}$ in 2006. These reported annual ET rates fall within our estimates in the present study that shows ET for plantation pine (1990s) having ET rate of 1000-1200 mm year$^{-1}$ (Fig. 17).

Monthly scale empirical ET models for various ecosystems have been developed (Sun et al., 2011; Fang et al., 2015) and the accuracy is generally uncertain due to the large variability of climate and ecosystem structure, and ET processes in general.  For example, Sun et al.(2011) proposed a general predictive model to estimate monthly ET using reference ET, precipitation and LAI over 13 ecosystems, which include the mid-rotation NC2 site and another clearcut site (NC1 in AmeriFlux database) also in our study area. The monthly ET estimates from that approach had a relative error of 23% and a RMSE of 15.1 mm month$^{-1}$, while STARFM relative errors from the current study were 13.6% at the monthly time step – a substantial improvement in accuracy.

## 5.3 Water use variations with stand age and land cover type

Previous studies investigating the relationship between land cover and age of forest stands on water usage mainly focused on the resulting impact on the water balance via impacts on streamflow at the watershed scale (Matheussen et al., 2000; Vertessy et al., 2001; Williams et al., 2012). Matheussen et al. (2000) analyzed the hydrological effects of land cover change in the Columbia River basin and found a significant correlation between hydrological change and  the tree maturity in the forested areas. In the current study, we found that water use increased linearly with stand age between 1 and 20 years, then plateaued or decreased with age about 20 years. An investigation of mountain ash forests in Victoria found that annual ET from a nearly 35-year stand was 245 mm more than that from 215-year stand (Vertessy et al., 2001). Similarly, another study of  three forest stands, aged 14, 45, and 160 years, found plot transpiration declined from 2.2 mm per day in the 14-year stand, 1.4mm per day in 45-year stand to 0.8mm per day in 160-year old forest (Roberts et al., 2001). In another study Kuczera (1987) found quickly rapid decrease of mean annual water yield from a mountain ash forest watershed when stand age increased from 1-year to around 25-year, suggesting greater water use by the older forest stands. Murakami et al. (2000) used a Penman-Monteith equation based model to simulate an ET- stand age relationship, which also showed a clear upward trend for young forest and a peak in ET at 20 years. The sharp increase of seasonal ET with the increase of stand age from 1 year to around 20 years illustrated in Fig. 17 is consistent with these earlier studies. Plant  LAI is closely related to ET (Sun et al., 2011) and is also an important input in plant physiological and hydrologic/land surface models as well as crop models (Duchemin et al., 2006; Nemani et al., 1993; Tague et al., 2012).  As shown in Fig. 16, we also find correlation between cumulative ET and season-average LAI over different land cover types, but LAI explained only 59% of the modeled variability in ET.



## 6 Conclusions

This study demonstrates the capability of the multi-scale data fusion ET model to estimate daily field-scale ET over a forested landscape. Daily ET retrievals over the growing season of 2013, generated at 30-m spatial resolution, compared well with observed fluxes at AmeriFlux tower sites in a mature pine stand and recent clear cut site, demonstrating capability to reasonably capture a range in land-surface conditions within a managed pine plantation. Errors were 29% at daily time step, 12% at monthly time step and 3% over the study period.

A new scene gap-filling method was described to maximize the number of Landsat images used for ET retrieval at Landsat scale, and will be of benefit in areas with persistent partial cloud cover and for recovering scenes from the Landsat 7 archive that are impacted by the SLC failure. The STARFM data fusion method can help to mitigate the dearth of high spatial-temporal resolution land surface temperature data from currently available satellite systems.

This study suggests that satellite retrievals of ET at the Landsat scale can be used to analyze water use variability over a heterogeneous forested landscape in response to stand age and vegetation composition. The estimates of ET at a high resolution provide insight of seasonal water balances and thus offers useful information for local water resource management. Comparing with traditional forest ET estimation methods, this study provides an accurate, efficient and more real-time estimation of landscape-level ET, which is ready and suitable for operational production.

**Acknowlegements** This work was funded in part by a grant from NASA (NNH14AX36I). We thank the Weyerhaeuser Company for providing stand age data. The U.S. Department of Agriculture (USDA) prohibits discrimination in all its programs and activities on the basis of race, color, national origin, age, disability, and where applicable, sex, marital status, familial status, parental status, religion, sexual orientation, genetic information, political beliefs, reprisal, or because all or part of an individual's income is derived from any public assistance program. (Not all prohibited bases apply to all programs.) Persons with disabilities who require alternative means for communication of program information (Braille, large print, audiotape, etc.) should contact USDA's TARGET Center at (202) 720-2600 (voice and TDD). To file a complaint of discrimination, write to



USDA, Director, Office of Civil Rights, 1400 Independence Avenue, S.W., Washington, D.C. 20250-9410, or call (800) 795-3272 (voice) or (202) 720-6382 (TDD). USDA is an equal opportunity provider and employer.

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



Table 1. Landsat imagery used in the study.

| Sensor | Landsat7 | Landsat8 | Landsat8 | Landsat8 | Landsat8 | Landsat8 | Landsat8 | Landsat8 |
|---|---|---|---|---|---|---|---|---|
| DOY | 96 | 104 | 136 | 152 | 200 | 248 | 312 | 328 |
| %Cloudiness | 0.1 | 0 | 16.4 | 5.1 | 23.8 | 0.3 | 0.1 | 0.1 |
| %SLC Gap | 37 | NA | NA | NA | NA | NA | NA | NA |

Table 2. Summary of the statistical indices quantifying model performance for instantaneous and daytime

integrated surface energy fluxes on Landsat overpass dates.

| NC2 Site | | | | | | | | | | | | |
|---|---|---|---|---|---|---|---|---|---|---|---|
| | Daily Fluxes | | | | | | Instantaneous Fluxes | | | | | |
| Variable | $R_sd$ | $R_nd$ | Gd | Hd | LEd (Closed) | LEd (Unclosed) | $R_s$ | $R_n$ | $G_0$ | H | LE (Closed) | LE (Unclosed) |
| Unit | $MJ\ m^{-2}d^{-1}$ | $MJ\ m^{-2}d^{-1}$ | $MJ\ m^{-2}d^{-1}$ | $MJ\ m^{-2}d^{-1}$ | $MJ\ m^{-2}d^{-1}$ | $MJ\ m^{-2}d^{-1}$ | $Wm^{-2}$ | $Wm^{-2}$ | $Wm^{-2}$ | $Wm^{-2}$ | $Wm^{-2}$ | $Wm^{-2}$ |
| n | 8 | 8 | 8 | 8 | 8 | 8 | 8 | 8 | 8 | 8 | 8 | 8 |
| Ō | 20.70 | 14.60 | 0.10 | 4.80 | 9.70 | 9.40 | 714 | 540 | 1 | 214 | 326 | 288 |
| MAE | 1.10 | 1.70 | 0.50 | 2.80 | 2.30 | 2.90 | 63 | 48 | 34 | 78 | 68 | 113 |
| RMSE | 4.20 | 2.80 | 0.20 | 4.80 | 3.00 | 5.70 | 78 | 51 | 36 | 89 | 89 | 125 |
| MBE | 1.60 | -0.30 | 0.50 | -1.90 | 1.10 | 1.40 | 63 | 21 | 34 | -55 | 41 | 91 |
| NC3 Site | | | | | | | | | | | | |
| Variable | $R_sd$ | $R_nd$ | Gd | Hd | LEd (Closed) | LEd (Unclosed) | $R_s$ | $R_n$ | $G_0$ | H | LE (Closed) | LE (Unclosed) |
| Unit | $MJ\ m^{-2}d^{-1}$ | $MJ\ m^{-2}d^{-1}$ | $MJ\ m^{-2}d^{-1}$ | $MJ\ m^{-2}d^{-1}$ | $MJ\ m^{-2}d^{-1}$ | $MJ\ m^{-2}d^{-1}$ | $Wm^{-2}$ | $Wm^{-2}$ | $Wm^{-2}$ | $Wm^{-2}$ | $Wm^{-2}$ | $Wm^{-2}$ |
| n | 8 | 8 | 8 | 8 | 8 | 8 | 8 | 8 | 0 | 8 | 0 | 8 |
| Ō | 22.30 | 13.80 | NA | 5.20 | NA | 6.50 | 776 | 507 | NA | 205 | NA | 209 |
| MAE | 1.50 | 1.90 | NA | 2.00 | NA | 1.70 | 28 | 41 | NA | 64 | NA | 60 |
| RMSE | 2.30 | 2.20 | NA | 2.70 | NA | 2.20 | 31 | 44 | NA | 75 | NA | 72 |
| MBE | 0.70 | 0.90 | NA | -0.10 | NA | 0.50 | 2 | 29 | NA | -11 | NA | 10 |

$R_sd$, daytime integrated solar radiation; $R_nd$, daytime integrated net radiation; Gd, daytime integrated soil flux; Hd, daytime integrated sensible heat; LEd, daytime integrated latent heat; $R_s$, $R_n$, $G_0$, H and LE are instantaneous fluxes; Closed indicates energy balance closure by residual, while Unclosed indicates that energy balance closure not imposed on the EC measurements. In addition: n, number of observations; Ō, mean measured flux; MAE, mean absolute error between the modeled and measured quantities; RMSE, root mean square error; MBE, mean bias error.





Table 3. Statistical metrics describing comparison of retrieved ET timeseries with tower observations at NC2 and NC3 at daily and monthly timesteps, as well as cumulative values over the study period (DOY 50-330).

| Site | NC2 Site | | | | NC3 Site | | | |
|---|---|---|---|---|---|---|---|---|
| Index | STARFM | Landsat Only | MODIS Smoothed | ALEXI | STARFM | Landsat Only | MODIS Smoothed | ALEXI |
| | Daily | | | | Daily | | | |
| n | 281 | 216 | 281 | 281 | 223 | 199 | 223 | 223 |
| MAE (mm d$^{-1}$) | 0.80 | 0.64 | 0.66 | 0.85 | 0.83 | 0.70 | 2.00 | 1.45 |
| RMSE (mm d$^{-1}$) | 0.99 | 0.83 | 0.85 | 1.04 | 1.02 | 0.88 | 2.27 | 1.84 |
| MBE (mm d$^{-1}$) | -0.03 | -0.29 | -0.10 | -0.11 | -0.05 | 0.10 | 1.96 | 1.31 |
| RE (%) | 27.9 | 19.2 | 22.0 | 28.0 | 30.6 | 25.1 | 85.0 | 56.0 |
| | Monthly | | | | Monthly | | | |
| n | 12 | 8 | 12 | 12 | 10 | 8 | 10 | 10 |
| MAE (mm d$^{-1}$) | 0.41 | 0.36 | 0.48 | 0.44 | 0.23 | 0.33 | 1.61 | 1.06 |
| RMSE (mm d$^{-1}$) | 0.53 | 0.39 | 0.58 | 0.54 | 0.29 | 0.36 | 1.82 | 1.33 |
| MBE (mm d$^{-1}$) | 0.00 | -0.25 | -0.11 | -0.15 | -0.01 | 0.16 | 1.61 | 1.06 |
| RE (%) | 13.6 | 12.1 | 16.1 | 14.8 | 10.9 | 15.5 | 76.3 | 50.1 |
| | Study Period | | | | Study Period | | | |
| n | 1 | 1 | 1 | 1 | 1 | 1 | 1 | 1 |
| MAE (mm d$^{-1}$) | 0.03 | 0.04 | 0.10 | 0.11 | 0.13 | 0.23 | 1.97 | 1.34 |
| MBE (mm d$^{-1}$) | -0.03 | 0.04 | -0.10 | -0.11 | 0.13 | 0.23 | 1.97 | 1.34 |
| RE (%) | 0.9 | 1.3 | 2.7 | 3.2 | 5.8 | 9.7 | 84.8 | 58.0 |

RE, relative error, which is calculated by dividing MAE by observed average ET.





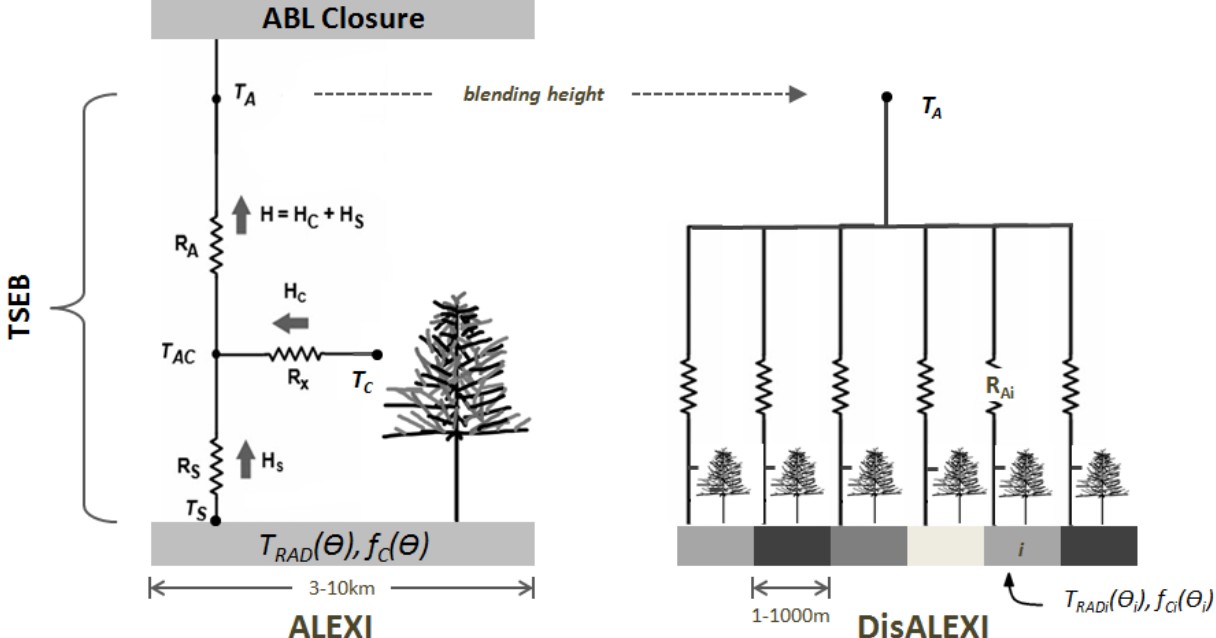

Figure 1. Schematic diagram of the ALEXI and DisALEXI modeling schemes. The left panel shows TSEB is employed to partition the income radiometric temperature ($T_{RAD}(\theta)$, $\theta$ is view angle) into vegetation (subscript "c") and soil (subscript "s") components based on vegetation coverage ($f(\theta)$). Sensible heat ($H$) is regulated by

5   the aerodynamic resistance ($R_a$), bulk leaf boundary layer resistance ($R_x$), and soil surface boundary layer resistance ($R_s$). ALEXI combines the TSEB and ABL model to estimate air temperature ($T_A$) at the blending height. The right panel represents the disaggregation of ALEXI output to finer scales based on LST and $f(\theta)$ information from Landsat and MODIS.





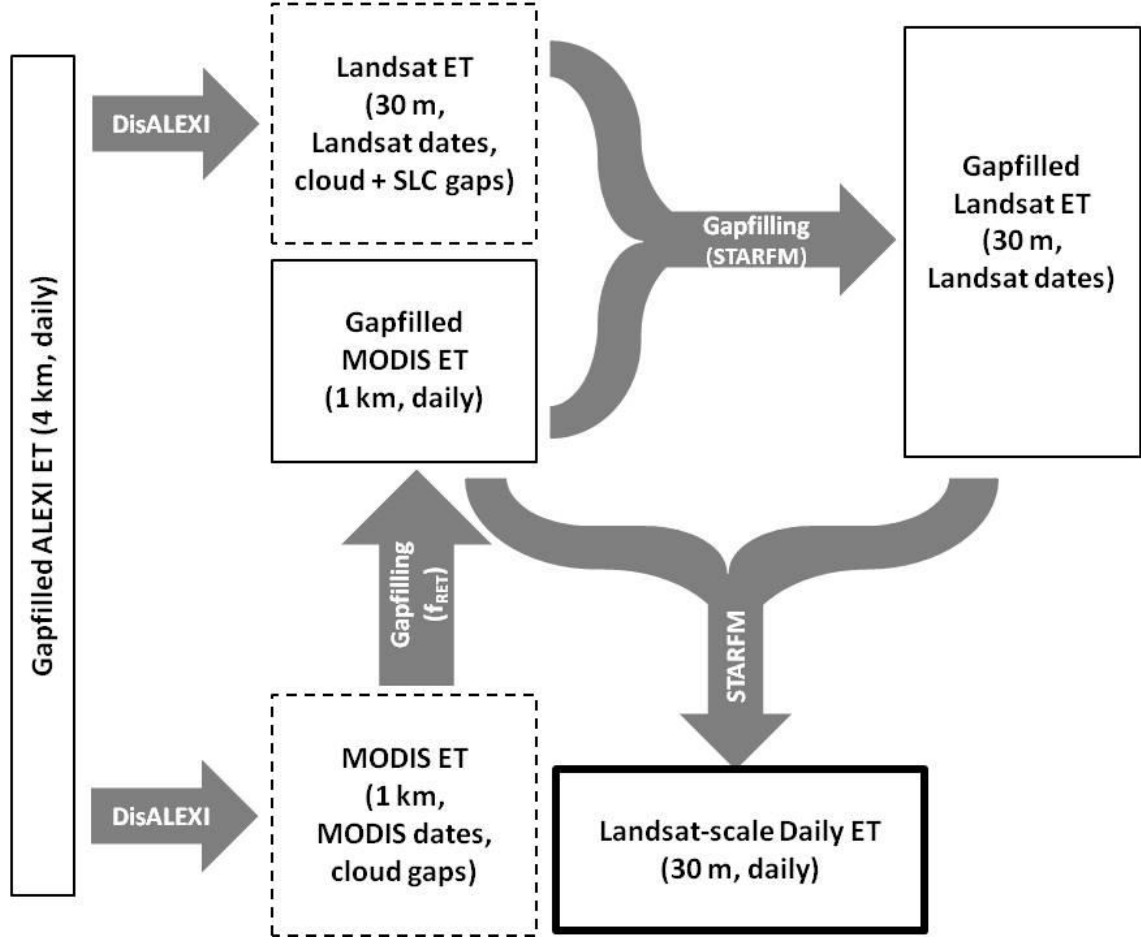

Figure 2. Flowchart describing the Landsat gap-filling and data fusion method. The arrows represent the methods applied. The boxes represent the datasets with different spatial and temporal characteristics created during the process. The dashed boxes indicate ET products with partially filled scenes (due to clouds or SLC gaps), solid boxes identify gap-filled scenes, and the thick box highlights the final gap-filled, 30-m daily product.





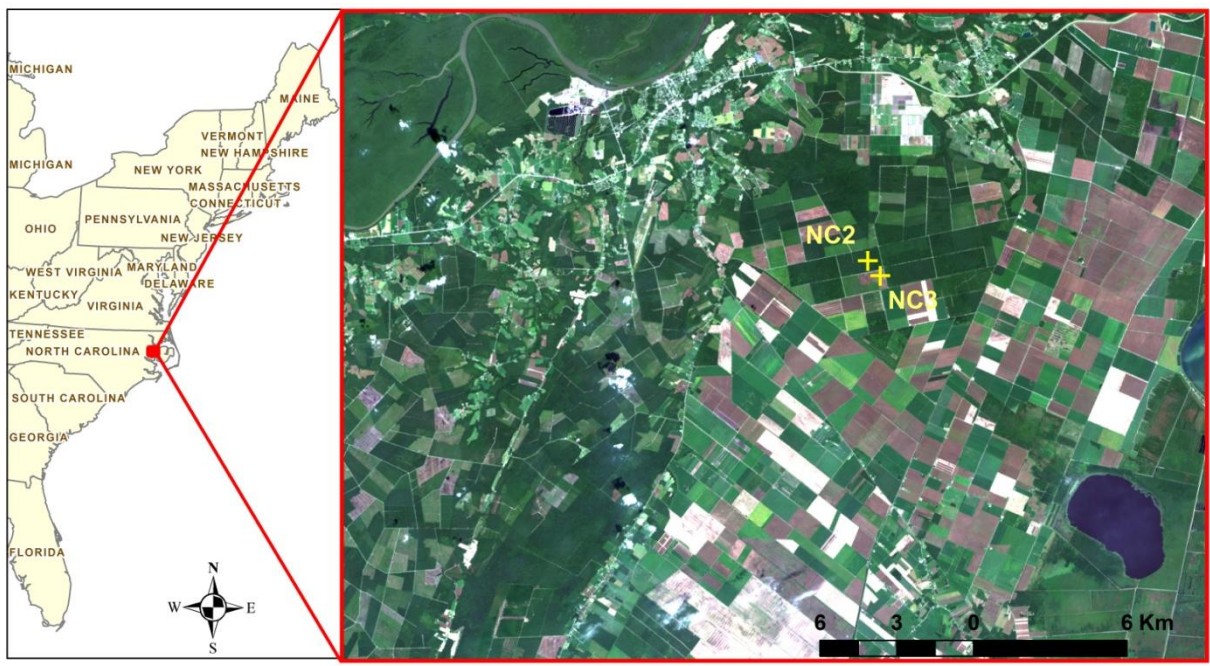

Figure 3. A Landsat 8 true color image (September 5$^{th}$, 2013) showing the North Caroline study area. The yellow crosses indicate the location of NC2 and NC3 flux towers.



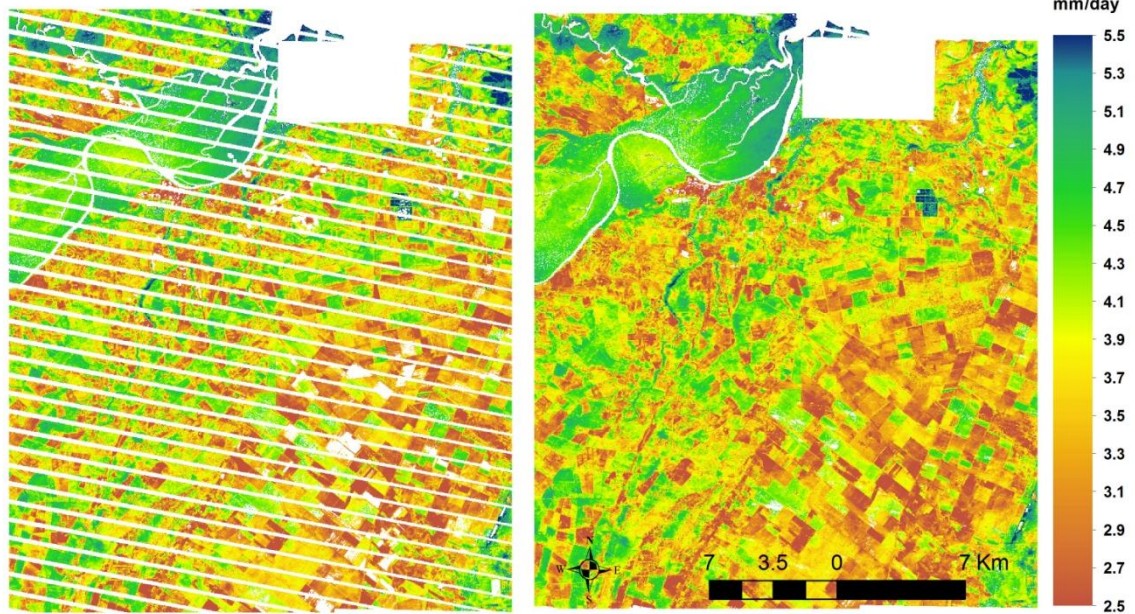

Figure 4. Example of gap-filling SLC-off stripes in a Landsat 7 ET image for DOY 96. The left image is Landsat retrieved ET with stripes and the right image is the filled ET with the Landsat gap-filling method.




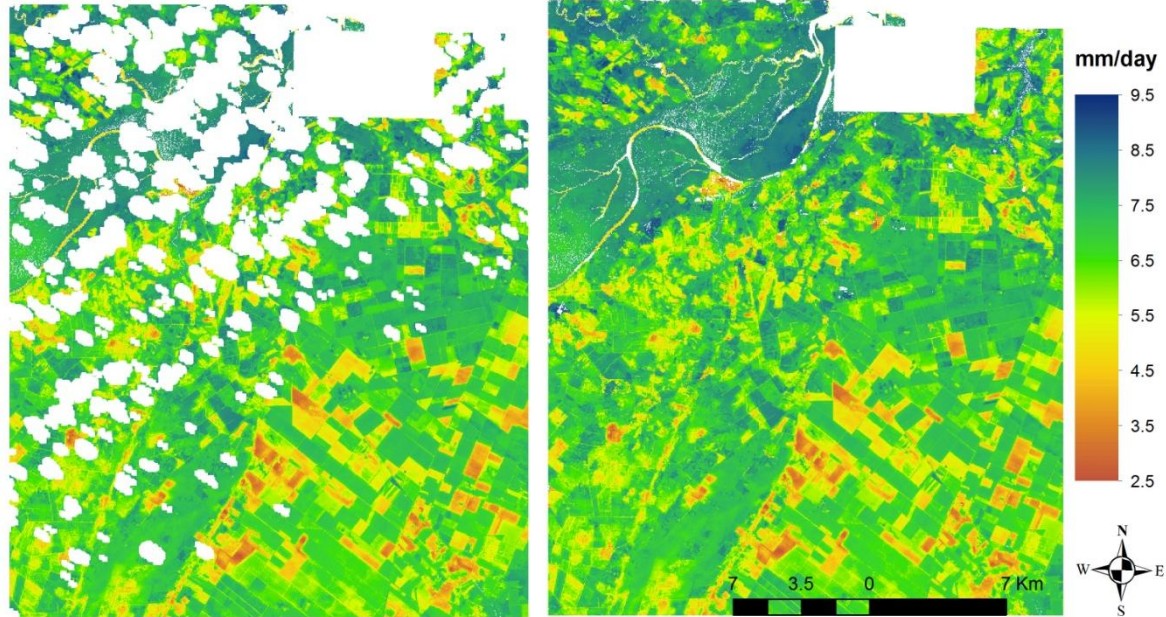

Figure 5. Example of gap-filling cloudy regions in a Landsat 8 ET image for DOY 200. The left image is Landsat-retrieved ET with clouds masked using the Fmask data layer and the right image is the filled ET with the Landsat gap-filling method.




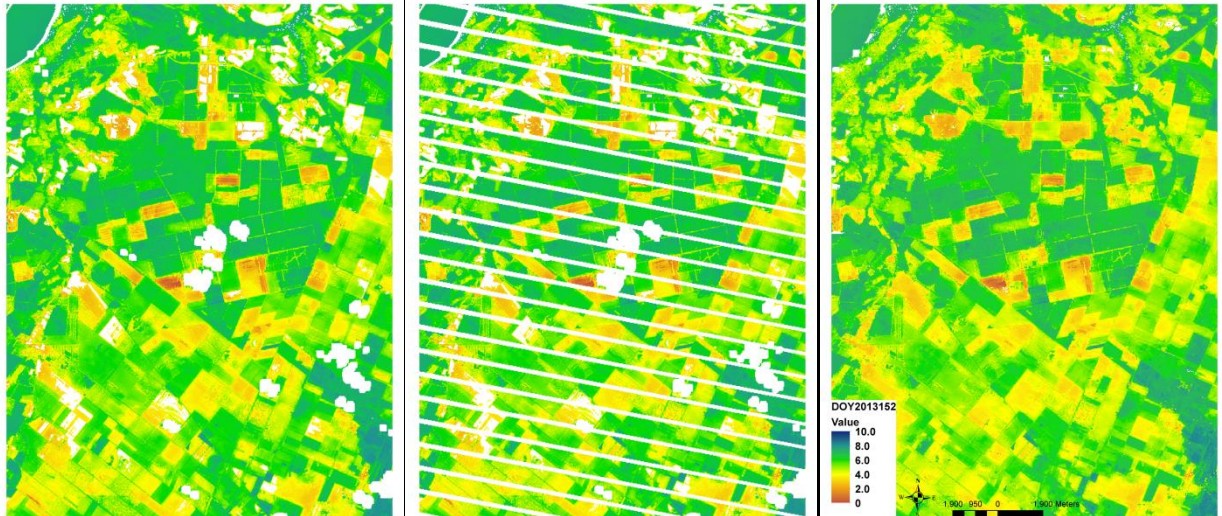

Figure 6. Comparison between the original Landsat ET retrieval for DOY 152 (left panel), an artificially gapped version, imposing SLC gaps from DOY 96 (middle panel), and the gap-filled map (right panel).



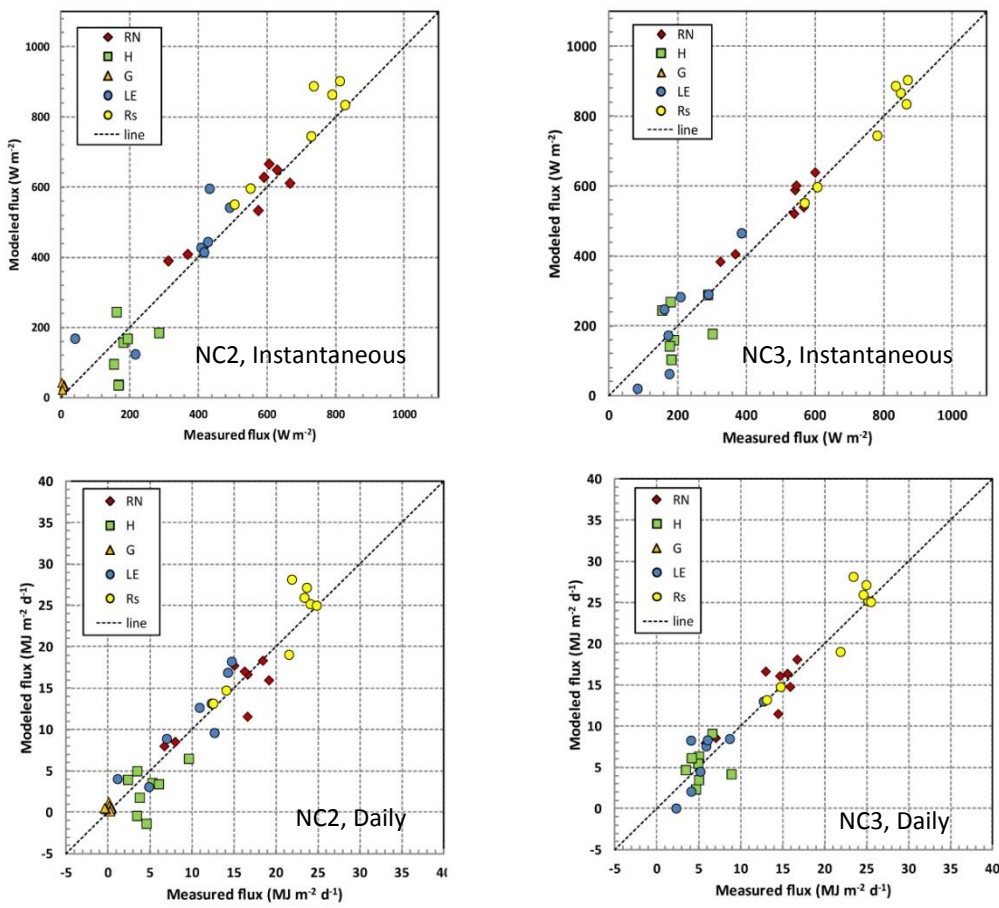

Figure 7. (left panel) Scatterplot of modeled and measured instantaneous and daily surface fluxes on Landsat overpass dates for NC2 flux tower sites. (right panel) Scatterplot of modeled and measured instantaneous and daily surface fluxes on Landsat overpass dates for NC3 flux tower sites.



Figure 8. Comparison of time series of ALEXI ET (4km), observed ET, Landsat ET retrieved on Landsat
overpass dates, Landsat-only interpolated ET and Landsat-MODIS fused ET for NC2 site (top panel) and NC3
site (bottom panel) in 2013.




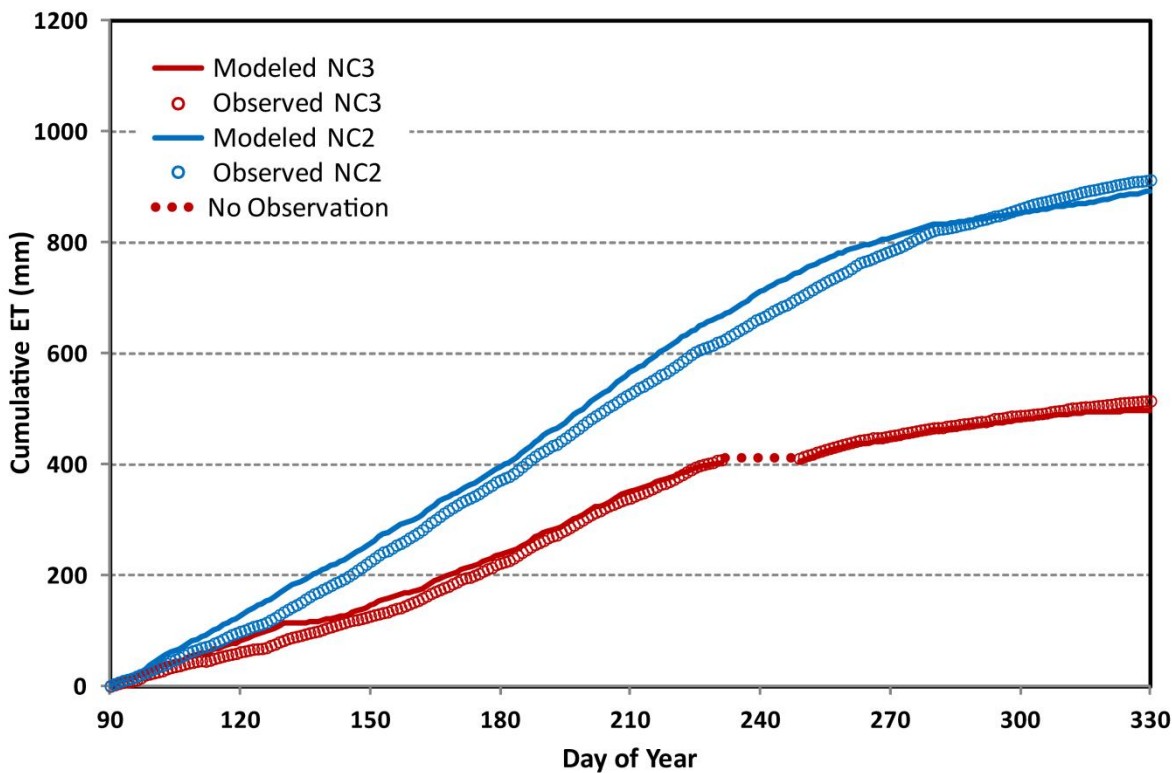

Figure 9. Comparison between the modeled and observed seasonal cumulative ET at NC2 and NC3 during 2013.





Figure 10. Land cover types over the study area from NLCD 2006. Area in the black outline is the plantation area.





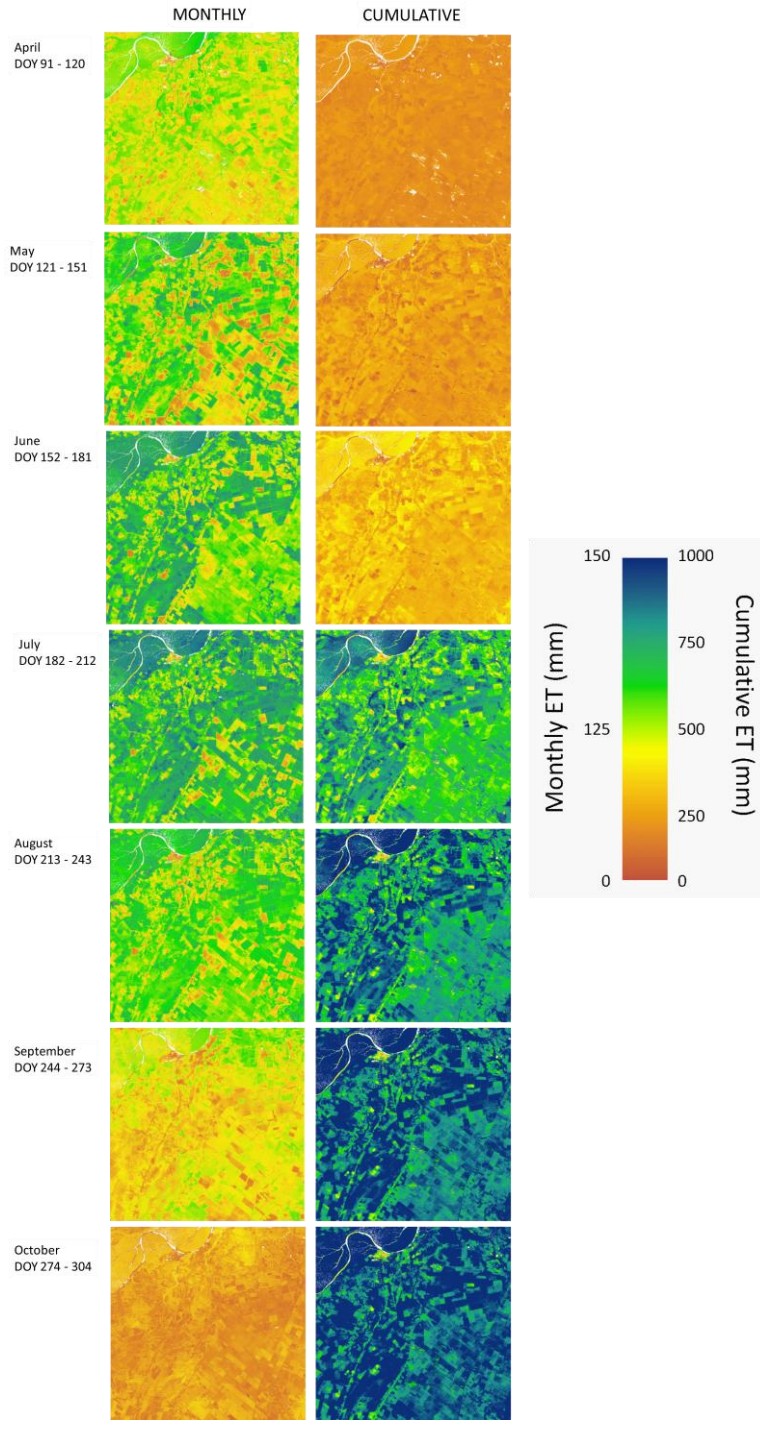

Figure 11. Spatial patterns of monthly cumulative ET (left column) from April to October and cumulative ET on the end day of each month over the study area.



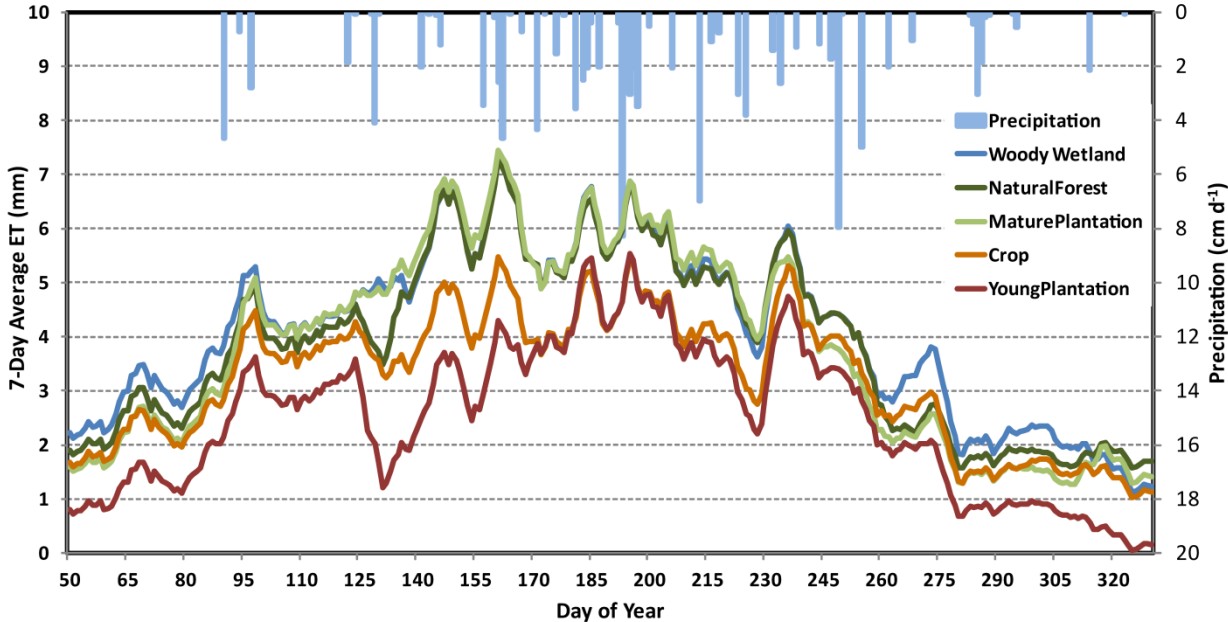

Figure 12. Time series of modeled plot-scale ET (daily values smoothed with a seven-day moving average) associated with different land cover types.





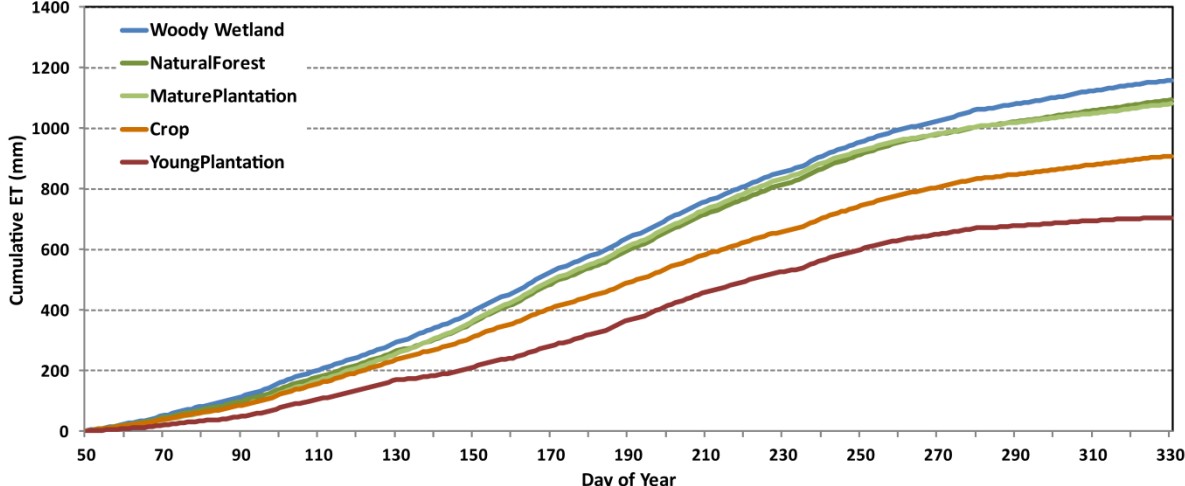

Figure 13. Seasonal cumulative ET for different land cover types.





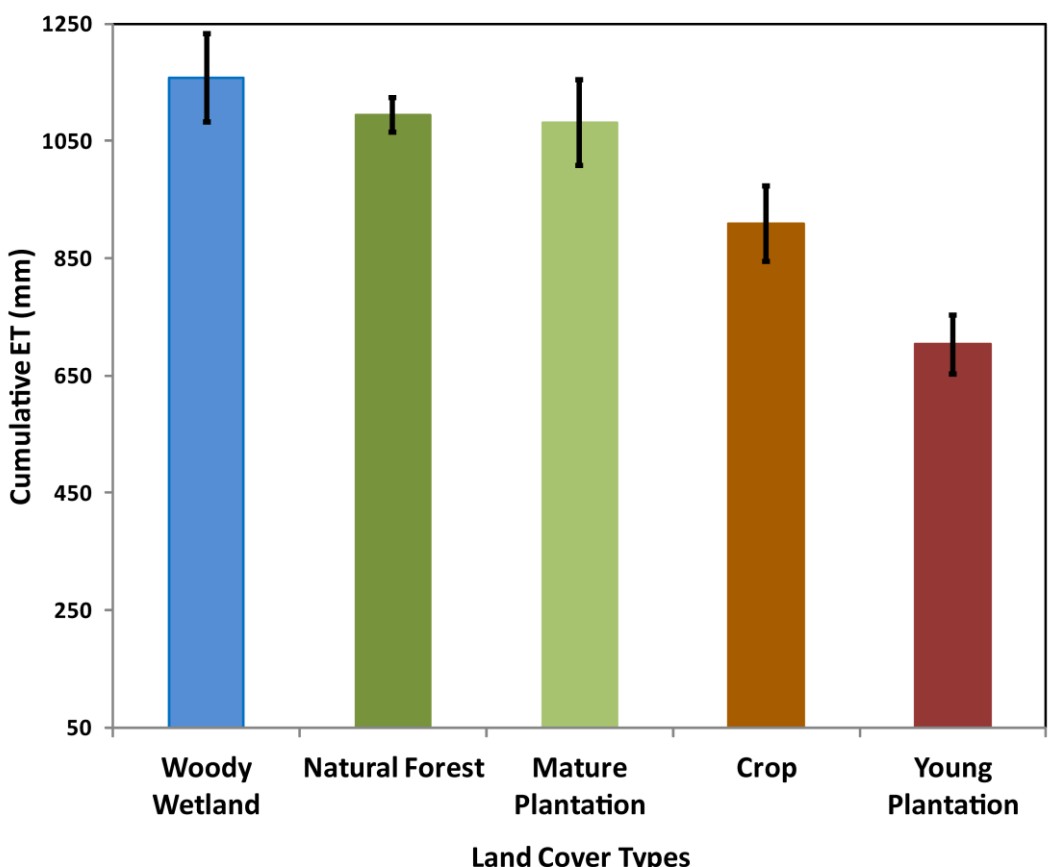

Figure 14. Average cumulative ET at DOY 330 in 2013 over different land cover types, and standard deviations within the sample populations (the black bar).



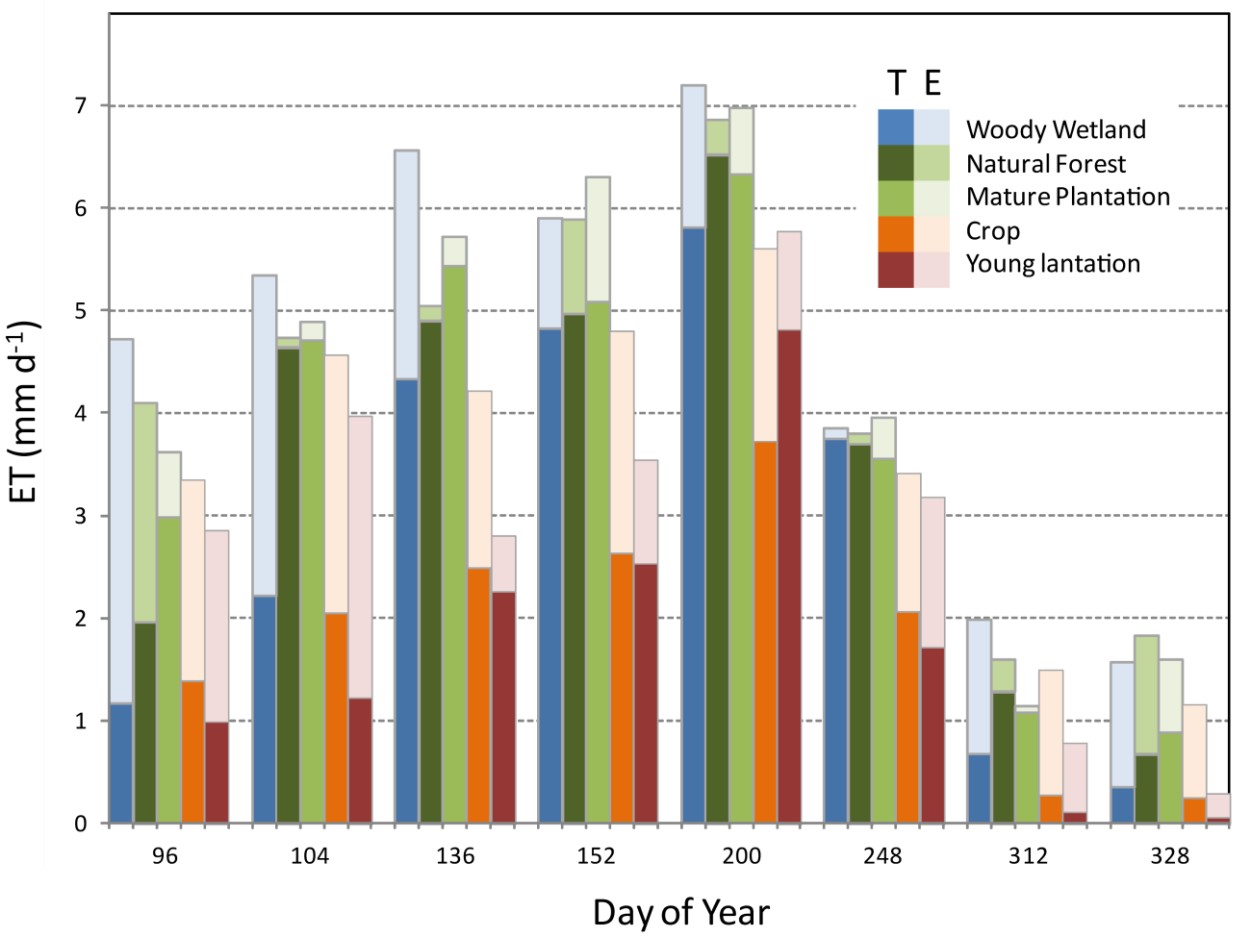

Figure 15. Average evaporation (E) and transpiration (T) components of ET for five land cover types on Landsat overpass days.




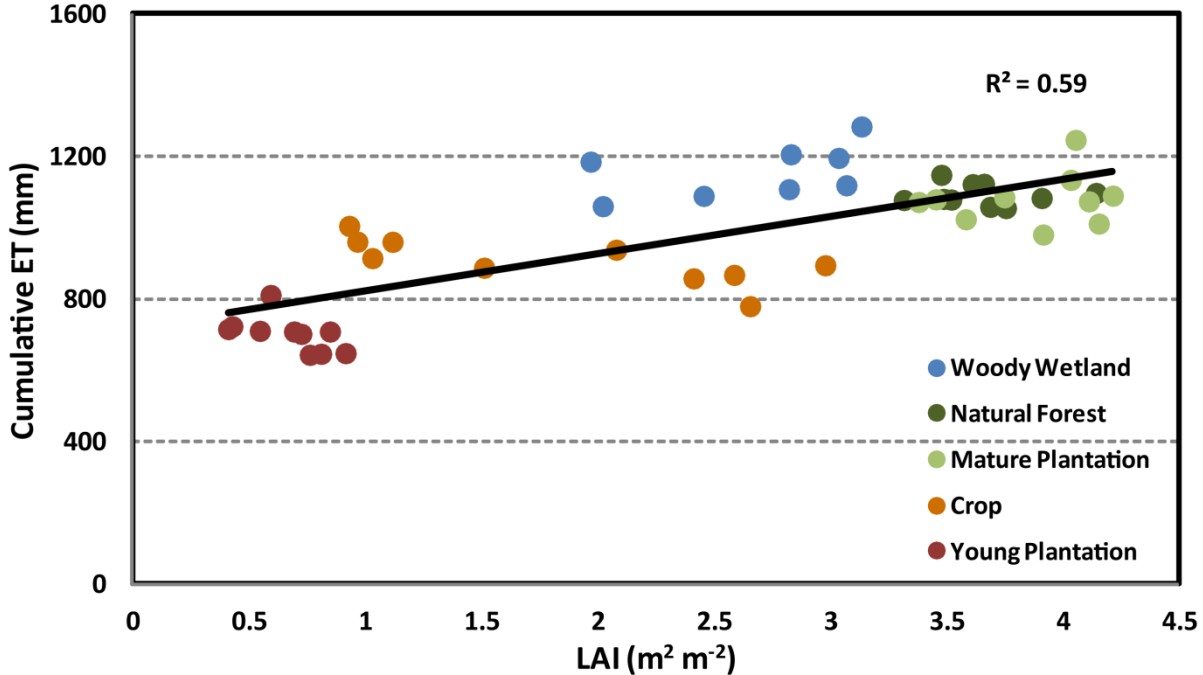

Figure 16. Modeled cumulative ET from DOY 50 to DOY 330 as a function of LAI.





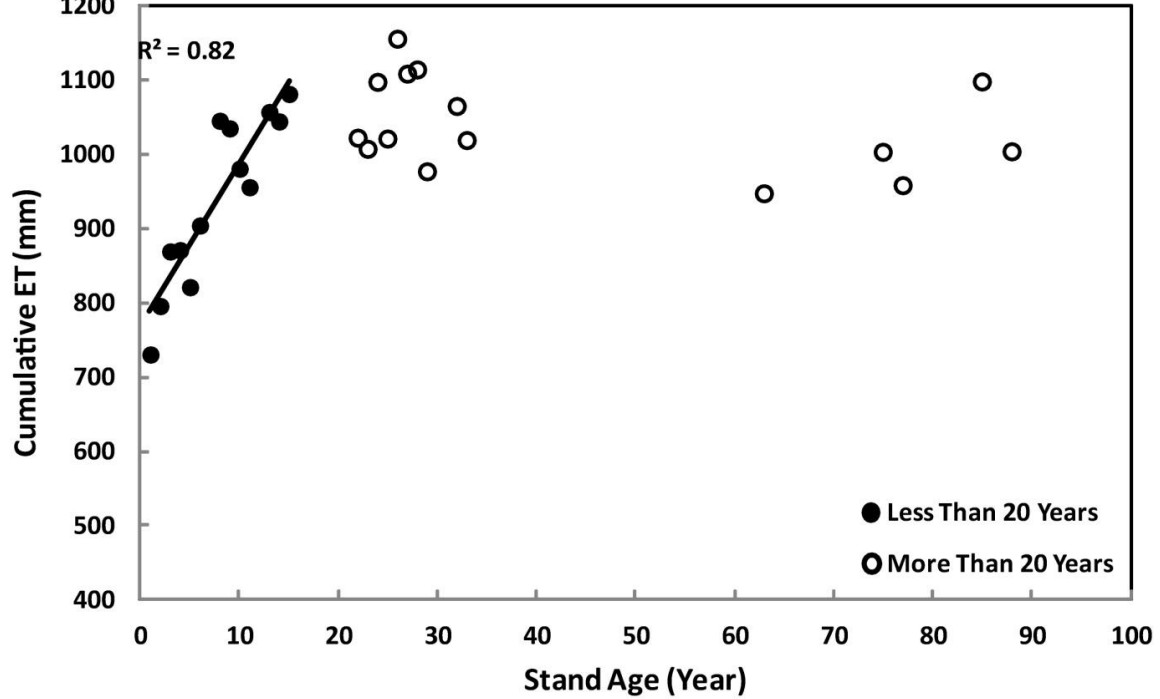

Figure 17. Modeled cumulative ET at DOY 330 as a function of stand age.