# Peer review of "Daily Landsat-scale evapotranspiration estimation over a forested landscape in North Carolina, USA using multi-satellite data fusion"

_Hydrology and Earth System Sciences, 2016_

## Referee Comment (RC1) · Anonymous Referee #1 · 30 Aug 2016

The manuscript "Daily Landsat-scale evapotranspiration estimation over a forested landscape in North Carolina, USA using multi-satellite data fusion" addresses the lack of spatial and temporal resolution when using satellite imagery to estimate ET over heterogeneous landscapes. The authors overcome this problem by the application of multi-sensor data fusion combined with the STARF model. Additionally, the authors introduce a new method, based on STARFM, to fill gaps in primary data sets caused, for example, by cloud cover. Their method is validated with empirical data from two eddy covariance flux towers. The study is novel and innovative, and well structured and written. Figures and tables are widely appropriate. I only have minor suggestions and comments to improve the manuscript's readability and consistency.

[Figure]

Specific comments: P2L14: ET also varies with different development stages, as actually demonstrated by the authors' own study. P3L11-12: "plant status" ... please be more specific, do you refer to the development stage? P3L18: Add von Bertalanffy (1968), "General Systems Theory" to the lists of references as he was one of the first addressing the equifinality problem. P4L18: What methodological challenges do the described differences between forest land cover and shorter crops cover imply for this study? P4L25-27: "We also present a new method, ..., for filling gaps ..." Sounds nearly redundant but is one of the primary novelty of the paper as far as I understand. Shouldn't it be more upfront then? Equations in general: Please add the units to the description of each parameter. Equation 2: It's not clear to me what the purpose is of presenting the general equation first and then the two equations specifically referring to canopy and soil. If redundant, remove the general version and tag the other two as (a) and (b). P5L22-23: Did you mean "... T is the air temperature measured at height $Z_T$ ..."? P10L26: Replace "3" with "three" P12L9: Same as above. "... including one Landsat 7 scene and seven from Landsat 8 ..." P13L13-21: I think this paragraph goes beyond the classic presentation of results and should be moved to the discussion section. P13L27: Put "ET" in brackets. P14L2: "... 3 site ..." Typo? P15L1-2: Inconsistency. "... ET was 3% of the total observed flux at NC2 and -4% at NC3" In Fig. 9, at both sites the modelled ET is below the observed ET. P15L7-9: "[Note: ....]" Please use a footnote instead. P19L18: Add "(Australia)" after "Victoria". Fig. 1, caption: For consistency, replace "vegetation" with "canopy". Fig. 7, caption: Repetitive. Condense. Fig. 7, legends: "1:1 line", "LE" should be "$\lambda E$". Figs. 13 and 14: Merge. The information is the same apart from the standard deviations in Fig. 14. Fig. 15, legend: Typo ... "Young Plantation"
* * *

---

## Referee Comment (RC2) · Anonymous Referee #2 · 8 Nov 2016

This manuscript describes the application of a multi-product (GOES, MODIS and Landsat) data fusion approach for estimating evapotranspiration (ET) at high spatial and temporal resolution in a heterogeneous landscape. ET is resolved with sufficient spatial resolution to determine differences in fields or plantations with different vegetation types or histories, and the model is validated against eddy covariance observations. A new approach for gap-filling Landsat scenes contaminated by clouds or the scan-line corrector failure is described.

In general, the manuscript is well written and clear, will be of interest to the HESS readership and represents a valuable contribution. Some comments regarding the manuscript are provided below to help with improving clarity.

**Specific comments**

**Methods**

The section on the parameterization of aerodynamic resistances, to which the expressions for the soil and leaf boundary resistances could also be added. When referring to Fig 1 (and in the fig caption) the reader can be referred to the supplementary information for the expressions for the resistance terms.

I appreciate that the experimental site and datasets are included in a single section, however, it might improve clarity and flow if the section concerning ALEXI/DisALEXI model inputs (3.3) were moved to be closer to the description of the models (e.g., immediately after section 2.1). There are important details in section 3.3 [e.g., parameterization of f(φ)] that would be better situated nearer the description of the models.

Was the storage flux estimated from profile measurements and used when computing net ecosystem water vapor exchange at the eddy covariance sites?

**Results/Discussion**

Were there any cases where the flux towers were obscured by clouds in Landsat scenes? If so, how, did the Landsat gap-filling perform under such circumstances?

What is the latency of the "finished" ET product (Landsat-like resolution)? How feasible would it be for using this for water management planning throughout the growing season?

---

## Author Comment (AC1) · 29 Nov 2016

Reviewer's report: hess-2016-198

 Reviewer #1:

The manuscript "Daily Landsat-scale evapotranspiration estimation over a forested landscape in North Carolina, USA using multi-satellite data fusion" addresses the lack of spatial and temporal resolution when using satellite imagery to estimate ET over heterogeneous landscapes. The authors overcome this problem by the application of multi-sensor data fusion combined with the STARF model. Additionally, the authors introduce a new method, based on STARFM, to fill gaps in primary data sets caused, for example, by cloud cover. Their method is validated with empirical data from two eddy covariance flux towers. The study is novel and innovative, and well structured and written. Figures and tables are widely appropriate. I only have minor suggestions and comments to improve the manuscript's readability and consistency.

We are greatly thankful to the reviewer for this thorough and thoughtful review.

Specific comments:

P2L14: ET also varies with different development stages, as actually demonstrated by the authors' own study.

Great point. We now added development stages in the sentence as one of the parameters that affect ET.

P3L11-12: "plant status" … please be more specific, do you refer to the development stage?

We changed "plant status" to "plant growth rate".

P3L18: Add von Bertalanffy (1968), "General Systems Theory" to the lists of references as he was one of the first addressing the equifinality problem.

Thanks for providing the citation, we added it in.

P4L18: What methodological challenges do the described differences between forest land cover and shorter crops cover imply for this study?

We added this statement:  "This presents a modeling challenge in terms of accurately defining turbulent exchange coefficients, as well as describing radiation transport through the canopy."

P4L25-27: "We also present a new method, …, for filling gaps …" Sounds nearly redundant but is one of the primary novelty of the paper as far as I understand. Shouldn't it be more upfront then?

We agree that we should put this new gap-filling method more upfront. We have rephrased this with a stronger statement.  We prefer to maintain the distinction between science objectives and this methodological advancement, however.

"Additionally, we present a novel methodological advancement, based on data fusion, for filling gaps in Landsat-based ET retrievals due to partial cloud cover as well as the scan-line corrector (SLC) failure in Landsat 7. This technique facilitates more complete use of the existing Landsat archive for investigating water use dynamics at the landscape scale. "

Equations in general: Please add the units to the description of each parameter.

We added the units to the description of model parameters and variables.

Equation 2: It's not clear to me what the purpose is of presenting the general equation first and then the two equations specifically referring to canopy and soil. If redundant, remove the general version and tag the other two as (a) and (b).

The three equations have been retained to signify that the model solves for both the component and system fluxes. However, we rearranged terms in the first equation to make the 3 more parallel.

P5L22-23: Did you mean "... T is the air temperature measured at height Z_T ..."?

$Z_T$ is a parameter in equation (3). This is now clarified within the text.

P10L26: Replace "3" with "three"

We removed "3" in this sentence, as it seemed redundant.

P12L9: Same as above. "... including one Landsat 7 scene and seven from Landsat 8 ..."

We replaced "1" with "one" and "7" with "seven".

P13L13-21: I think this paragraph goes beyond the classic presentation of results and should be moved to the discussion section.

We agree with the reviewer that this material did not belong in results. The contrast with previous attempts to gap-fill Landsat 7 imagery is now made within the methods section.

P13L27: Put "ET" in brackets.

Reference to ET was removed here, because Table 2 refers only to energy fluxes.

P14L2: "... 3 site ..." Typo?

The typo was removed.

P15L1-2: Inconsistency. "... ET was 3% of the total observed flux at NC2 and -4% at NC3" In Fig. 9, at both sites the modelled ET is below the observed ET.

We checked the data and fixed the typo.

P15L7-9: "[Note: ....]" Please use a footnote instead.

*It is now inserted as the footnote.*

P19L18: Add "(Australia)" after "Victoria".

*Australia was added.*

Fig. 1, caption: For consistency, replace "vegetation" with "canopy".

*We replaced "vegetation" with "canopy".*

Fig. 7, caption: Repetitive. Condense.

*We modified the caption to condense it.*

Fig. 7, legends: "1:1 line", "LE" should be "\lambdaE".

*Legends were fixed.*

Figs. 13 and 14: Merge. The information is the same apart from the standard deviations in Fig. 14.

*While there is some overlap in information, we believe there is visual value in retaining the bar chart with the standard deviations to more clearly visualize differences in the total seasonal water use, as well as the variability across the scene. The cumulative curves provide temporal information, and would become too cluttered if the range in variation was superimposed.*

Fig. 15, legend:
Typo ... "Young Plantation"

*Typo was fixed.*

---

## Author Comment (AC2) · 29 Nov 2016

Reviewer #2:

This manuscript describes the application of a multi-product (GOES, MODIS and Landsat) data fusion approach for estimating evapotranspiration (ET) at high spatial and temporal resolution in a heterogeneous landscape. ET is resolved with sufficient spatial resolution to determine differences in fields or plantations with different vegetation types or histories, and the model is validated against eddy covariance observations. A new approach for gap-filling Landsat scenes contaminated by clouds or the scan-line corrector failure is described.
In general, the manuscript is well written and clear, will be of interest to the HESS readership and represents a valuable contribution. Some comments regarding the manuscript are provided below to help with improving clarity.

Thank you for the review. We appreciate the feedback.

**Specific comments**
**Methods**
The section on the parameterization of aerodynamic resistances, to which the expressions for the soil and leaf boundary resistances could also be added. When referring to Fig 1 (and in the fig caption) the reader can be referred to the supplementary information for the expressions for the resistance terms.

We agree that the expressions for the soil and leaf boundary resistances could also be included. Since there are already papers listed these equations in details, we referred the readers to the citation.

I appreciate that the experimental site and datasets are included in a single section, however, it might improve clarity and flow if the section concerning ALEXI/DisALEXI model inputs (3.3) were moved to be closer to the description of the models (e.g., immediately after section 2.1). There are important details in section 3.3 [e.g., parameterization of f(φ)] that would be better situated nearer the description of the models.

We prefer to keep the details regarding specific sources of model inputs separate from the discussion of the models themselves. However, we did add a reference to the use of Beer's Law to compute f(φ) from LAI in Sec 2.1.

Was the storage flux estimated from profile measurements and used when computing net ecosystem water vapor exchange at the eddy covariance sites?

It was not bias-corrected using the energy balance method, so no consideration was made regarding change in storage of heat within the forest profile. Since we are mostly focused on daily timescale, the impact should be small.

**Results/Discussion**
Were there any cases where the flux towers were obscured by clouds in Landsat scenes? If so, how, did the Landsat gap-filling perform under such circumstances?

We did not use scenes that were largely cloudy – these tend to be difficult to fill and thoroughly cloud-clear.  The remaining scenes were clear over the tower sites.

What is the latency of the "finished" ET product (Landsat-like resolution)? How feasible would it be for using this for water management planning throughout the growing season?

This is a good question.  We have added a paragraph to Section 5.1 under Discussion:
"For real-time applications in water management, STARFM can be used to project water-use information beyond the date of the last Landsat overpass within some limited time-range, assuming MODIS data are available with low time latency.  Practical limits to a viable projection time range may vary with site and season, and will depend on the rate of change in weighting factors governing the STARFM fusion process."

---

## Author Comment (AC3) · 29 Nov 2016

[revised manuscript text omitted]
 and the right image is the filled EThas been gap-filled using with the Landsat gap-filling methodthe method described in Sec. 2.2.3.

[Figure]

Figure 5. Example of gap-filling cloudy regions in a Landsat 8 ET image for DOY 200. The left image is Landsat-retrieved ET with clouds masked using the Fmask data layer and the right image has been processed through the Landsat gap-filling method.

[Figure]

Figure 6. Comparison between the original Landsat ET retrieval for DOY 152 (left panel), an artificially gapped version, imposing SLC gaps from DOY 96 (middle panel), and the gap-filled map (right panel).

[Figure]

Figure 7. (left panel) Scatterplot of modeled and measured instantaneous (top row) and daily surface fluxes (bottom row) on Landsat overpass dates for NC2 (left column)flux tower sites. (right panel)and NC3 (right column) flux tower sites. Scatterplot of modeled and measured instantaneous and daily surface fluxes on Landsat overpass dates for NC3 flux tower sites.

[Figure]

[Figure]

Figure 8. Comparison of time series of ALEXI ET (4km), observed ET, Landsat ET retrieved on Landsat overpass dates, Landsat-only interpolated ET and Landsat-MODIS fused ET for the NC2 site (top panel) and NC3 site (bottom panel) sites in 2013.

[Figure]

Figure 9. Comparison between the modeled and observed seasonal cumulative ET at NC2 and NC3 during 2013.

[Figure]

Figure 10. Land cover types over the study area from NLCD 2006. Area in the black outline is the plantation area.

[Figure]

Figure 11. Spatial patterns of monthly cumulative ET (left column) from April to October and cumulative ET on the end day of each month over the study area.

[Figure]

Figure 12. Time series of modeled plot-scale ET (daily values smoothed with a seven-day moving average) associated with different land cover types.

[Figure]

Figure 13. Seasonal cumulative ET for different land cover types.

[Figure]

Figure 14. Average cumulative ET at DOY 330 in 2013 over different land cover types, and standard deviations within the sample populations (the black bar).

[Figure]

[Figure]

Figure 15. Average evaporation (E) and transpiration (T) components of ET for five land cover types on Landsat overpass days.

[Figure]

Figure 16. Modeled cumulative ET from DOY 50 to DOY 330 as a function of LAI.

[Figure]

Figure 17. Modeled cumulative ET at DOY 330 as a function of stand age.